# HAPI: A Large-scale Longitudinal Dataset of Commercial ML API Predictions

**Lingjiao Chen**[1], **Zhihua Jin**[2], **Sabri Eyuboglu**[1], **Christopher Ré**[1], **Matei Zaharia**[1], **James Zou**[1]

Stanford University[1], Hong Kong University of Science and Technology[2]

## Abstract

Commercial ML APIs offered by providers such as Google, Amazon and Microsoft have dramatically simplified ML adoption in many applications. Numerous companies and academics pay to use ML APIs for tasks such as object detection, OCR and sentiment analysis. Different ML APIs tackling the same task can have very heterogeneous performance. Moreover, the ML models underlying the APIs also evolve over time. As ML APIs rapidly become a valuable marketplace and a widespread way to consume machine learning, it is critical to systematically study and compare different APIs with each other and to characterize how APIs change over time. However, this topic is currently underexplored due to the lack of data. In this paper, we present HAPI (History of APIs), a longitudinal dataset of 1,761,417 instances of commercial ML API applications (involving APIs from Amazon, Google, IBM, Microsoft and other providers) across diverse tasks including image tagging, speech recognition and text mining from 2020 to 2022. Each instance consists of a query input for an API (e.g., an image or text) along with the API's output prediction/annotation and confidence scores. HAPI is the first large-scale dataset of ML API usages and is a unique resource for studying ML-as-a-service (MLaaS). As examples of the types of analyses that HAPI enables, we show that ML APIs' performance change substantially over time—several APIs' accuracies dropped on specific benchmark datasets. Even when the API's aggregate performance stays steady, its error modes can shift across different subtypes of data between 2020 and 2022. Such changes can substantially impact the entire analytics pipelines that use some ML API as a component. We further use HAPI to study commercial APIs' performance disparities across demographic subgroups over time. HAPI can stimulate more research in the growing field of MLaaS.

## 1 Introduction

Machine learning (ML) prediction APIs have dramatically simplified ML adoption. For example, one can use the Google speech API to transform an utterance to a text paragraph, or the Microsoft vision API to recognize all objects in an image. The ML-as-a-Service (MLaaS) market powered by these APIs is increasingly growing and expected to exceed $16 billion USD in the next five years [1].

Despite its increasing popularity, systematic analysis of this MLaaS ecosystem is limited, and many phenomena are not well understood. For example, APIs from different providers can have heterogeneous performance on the same dataset. Deciding which API or combination of APIs to use on a specific dataset can be challenging. Moreover, providers can update their ML APIs due to new data availability and model architecture advancements, but users often do not know how the API's behavior on their data changes. Such API shifts can substantially affect (and hurt) the performance of downstream applications. Certain biases or stereotypes in the ML APIs [30] can also be amplified or mitigated by API shifts. Understanding the dynamics of ML APIs is critical for ensuring the reliability of the entire user pipeline, for which the API is one component. It also helps users to adjust their API usage strategies timely and appropriately. For example, one may trust a speech API's

36th Conference on Neural Information Processing Systems (NeurIPS 2022) Track on Datasets and Benchmarks.

prediction if its confidence score is higher than 90% and invoke a human expert otherwise. Suppose the API is updated so that its confidence is reduced by 10% while its prediction remains the same (this happens in practice, as we will show). Then the human invocation threshold also needs to be adjusted to ensure consistent overall performance and human workload.

**Our contributions** In this paper, we present HAPI (History of APIs), a longitudinal dataset of 1,761,417 data points annotated by a range of different ML APIs from Google, Microsoft, Amazon and other providers from 2020 to 2022. This covers ML APIs for both standard classification such as sentiment analysis and structured prediction tasks including multi-label image classification. We have released our dataset on the project website [1], and will keep updating it by querying all ML APIs every few months. To the best of our knowledge, HAPI is the first systematic dataset of ML API applications. It is a unique resource that facilitates studies of the increasingly critical MLaaS ecosystem. Furthermore, we use HAPI to characterize interesting findings on API shifts between 2020 and 2022. Our analysis shows that API shifts are common: more than 60% of the 63 evaluated API-dataset pairs encounter performance shifts. Those API shifts lead to both accuracy improvements and drops. For example, Google vision API's shift from 2020 to 2022 brings a 1% performance drop on the PASCAL dataset but a 3.7% improvement on the MIR data. Interestingly, the fraction of changed predictions is often larger than the accuracy change, indicating that an API update may fix certain mistakes but introduce additional errors. ML APIs' confidence scores can also change even if the predictions do not. For example, from 2020 to 2021, the average confidence score of the Microsoft speech API increased by 30% while its accuracy became lower; in contrast, IBM API's confidence dropped by 1% but its accuracy actually improved. We also observe that subgroup performance disparity produced by different ML APIs is consistent over time. HAPI provides a rich resource to stimulate research on the under-explored but increasingly important topic of MLaaS.

## 2 Related Work

To the best of our knowledge, HAPI is the first large-scale ML API dataset. We discuss relevant literature below.

**MLaaS.** MLaaS APIs [36] have been developed and sold by giant companies including Google [9] and Amazon [2] as well as startups such as Face++ [6] and EPixel [5]. Many applications have been discovered [30, 50, 64], and prior work on ML APIs has spanned on their robustness [49] and pricing mechanisms [32]. One challenge in MLaaS is to determine which API or combination of them to use given a user budget constraint. This requires adaptive API calling strategies to jointly consider performance and cost, studied in recent work such as FrugalML [36] and FrugalMCT [33]. While they also released datasets of ML API predictions, their dataset only contain evaluation in one year. Recent work on API shift estimation [35] evaluated a few classification ML APIs in two years. On the other hand, HAPI provides a systematical evaluation of a number of ML APIs over a couple of years and thus enables more research on ML APIs evolution over time.

**Dynamics of ML systems.** ML is a fast growing community [66] and the update of one component may impact an ML system significantly. For example, a recent study on dataset dynamics [55] implies a concentration on fewer and fewer datasets over time and thus potentially increasing biases in many ML systems. Various hardware optimization [68] are shown to accelerate training and inference speeds for many applications. HAPI focuses on the dynamics of ML APIs, another important component of many ML systems.

**ML pipeline monitoring and assessments.** Monitoring and assessing ML pipelines are critical in real world ML applications. Existing work studies on how to estimate the performance of a deployed ML model based on certain statistics such as confidence [47], rotation prediction [40] and feature statistics of the datasets sampled from a meta-dataset [41]. More general approaches exploit human knowledge [37], white-box access to the ML models [31], or varying assumptions on label or feature distribution shifts [34, 38, 42]. Another line of work is identifying errors made by an ML model. This involves ML models for tabular data [29, 26] as well as multimedia data [54]. One common assumption made by them is that the deployed ML models are fixed and the performance change or error emergence is due to data distribution shifts. However, our analysis on HAPI indicates that ML

---

[1] http://hapi.stanford.edu/

systems powered by ML APIs may also change notably. This calls for monitoring and assessments under both model and data distribution shifts.

# 3 Construction of HAPI: Tasks, Datasets, and ML APIs

Table 1: Evaluated ML APIs. For each task, we have evaluated three popular ML APIs from different commercial providers. The valuation was conducted in the spring of 2020, 2021, and 2022 for classification tasks, and 2020 fall as well as 2022 spring for structured prediction tasks.

| Task Type | Task | ML API | | | Evaluation Period |
|---|---|---|---|---|---|
| Classify | SCR | Google [8] | Microsoft [14] | IBM [11] | March 2020, April 2021, May 2022 |
| | SA | Google [7] | Amazon [2] | Baidu [3] | March 2020, Feb 2021, May 2022 |
| | FER | Google [9] | Microsoft [13] | Face++ [6] | March 2020, Feb 2021, May 2022 |
| Struc Pred | MIC | Google [9] | Microsoft [13] | EPixel [5] | October 2020, Feb 2022 |
| | STR | Google [9] | iFLYTEK [12] | Tencent [19] | September 2020, March 2022 |
| | NER | Google [7] | Amazon [2] | IBM [10] | September 2020, March 2022 |

Table 2: Prices of ML services used for each task at their evaluation times. Price unit: USD/10,000 queries. We documented the price in 2020, 2021, and 2022 for standard classification tasks and 2020 and 2022 for structured predictions. Note that for the same task, the prices of different ML APIs are diverse. On the other hand, for a fixed ML API, its price is often stable over the past few years.

| Task | ML API | Price | | | ML API | Price | | | ML API | Price | | |
|---|---|---|---|---|---|---|---|---|---|---|---|---|
| | | 2020 | 2021 | 2022 | | 2020 | 2021 | 2022 | | 2020 | 2021 | 2022 |
| SCR | Google | 60 | 60 | 60 | MS | 41 | 41 | 41 | IBM | 25 | 25 | 25 |
| SA | Google | 2.5 | 2.5 | 2.5 | Amazon | 0.75 | 0.75 | 0.75 | Baidu | 3.5 | 3.6 | 3.7 |
| FER | Google | 15 | 15 | 15 | MS | 10 | 10 | 10 | Face++ | 5 | 5 | 5 |
| MIC | Google | 15 | | 15 | MS | 10 | | 10 | EPixel | 6 | | 6 |
| STR | Google | 15 | | 15 | iFLYTEK | 50 | | 52 | Tencent | 210 | | 210 |
| NER | Google | 10 | | 10 | Amazon | 3 | | 3 | IBM | 30 | | 30 |

Let us first introduce HAPI, a longitudinal dataset for ML prediction APIs. To assess ML APIs comprehensively, we designed HAPI to include evaluations of (i) a large set of popular commercial ML APIs for (ii) diverse tasks (iii) on a range of standard benchmark datasets (iv) across multiple years. For (ii), we consider six different tasks in two categories: standard classification tasks including spoken command recognition (SCR), sentiment analysis (SA), and facial emotion recognition (FER), and structured predictions including multi-label image classification (MIC), scene text recognition (STR), and named entity recognition (NER). To achieve (i) and (iv), we have evaluated three different APIs from leading companies for each task from 2020 to 2022, summarized in Table 1. Specifically, we have evaluated all classification APIs in the spring of 2020, 2021, and 2022, separately, and all structured prediction APIs in 2020 fall and 2022 spring respectively. The prices of all evaluated ML APIs are presented in Table 2. Note that, for any fixed task, the prices of different ML APIs vary in a large range. This implies selection of different ML APIs may impact the dollar cost of a downstream application. Interestingly, for a fixed ML API, there is almost no change in its price over the past few years. We will also continuously evaluate those APIs and update HAPI in the future.

What remains is on which datasets the ML APIs have been evaluated. To ensure (iii), we choose four commonly-used benchmark datasets for each classification task, and three datasets for each structure prediction task. The dataset statistics are summarized in Table 3. Note that those datasets are diverse in their size and number of labels, and thus we hope they can represent a large range of real world ML API use cases. Some datasets come with additional meta data. For example, the speaker accents are available for the spoken command dataset DIGIT. Such information can be used to study how an ML API's bias changes over time. We leave more details in the appendix.

Table 3: Datasets used to evaluate classification APIs (in tasks SCR, SA, FER) and structured prediction APIs (in tasks MIC, STR, NER). We queried each dataset on all three APIs that are relevant for that task.

| Task | Dataset | Size | # Labels | Dataset | Size | # Labels |
|---|---|---|---|---|---|---|
| Speech Command Recog | DIGIT [4] | 2000 | 10 | AMNIST [27] | 30000 | 10 |
| | CMD [70] | 64727 | 31 | FLUENT [59] | 30043 | 31 |
| Sentiment Analysis | IMDB [60] | 25000 | 2 | YELP [21] | 20000 | 2 |
| | WAIMAI [20] | 11987 | 2 | SHOP [16] | 62774 | 2 |
| Facial Emotion Recog | FER+ [25] | 6358 | 7 | RAFDB [56] | 15339 | 7 |
| | EXPW [72] | 31510 | 7 | AFNET [61] | 287401 | 7 |
| Multi-label Image Class | PASCAL [44] | 11540 | 20 | MIR [51] | 25000 | 25 |
| | COCO [57] | 123287 | 80 | | | |
| Scene Text Recog | MTWI [48] | 9742 | 4404 | ReCTS [71] | 20000 | 4134 |
| | LSVT [67] | 30000 | 4852 | | | |
| Named Entity Recog | CONLL [65] | 10898 | 9910 | GMB [28] | 47830 | 14376 |
| | ZHNER [22] | 16915 | 4375 | | | |

The output formats of different ML APIs are often different. For example, Google API generates a Google client object for each input data while Everypixel API simply returns a dictionary. To mitigate such heterogeneity, we propose a simple abstraction to represent an ML API's output. given each data point $x$ and evaluation time $t$, a classification ML API's output is (i) a predicted label $f(x, t)$ and (ii) the associated confidence score $q(x, t)$. For structured prediction tasks, the output includes (i) a set of predicted labels $f(x, t)$ (ii) associated with their quality scores $q(x, t)$. For each ML API and dataset pair, we recorded the API's prediction $f(x, t)$ and $q(x, t)$ at each evaluation time. We also include the true label $y$ for each dataset.

As a result, HAPI consists of 1,761,417 data samples from various tasks and datasets annotated by commercial ML APIs from 2020 to 2022. We provide download access on the project website, and also offer a few interesting examples for exploration purposes.

## 4   Example Analyses Enabled by HAPI: Model Shifts Over Time

We demonstrate the utility of HAPI by showing interesting insights that we can learn from it regarding how APIs change over time. The analysis here is not meant to be exhaustive; indeed we leave many open directions of investigation and encourage the community to dive deeper using HAPI. Our preliminary analysis goal is four-fold: (i) assess whether an ML API's predictions change over time, (ii) quantify how much accuracy improvements or declines are incurred due to ML API shifts, (iii) estimate to which direction prediction confidences of the ML APIs move, and (iv) understand how an ML API's gender and race biases evolve.

### 4.1   Findings on Classification APIs

We first study the shifts of ML APIs designed for simple classification tasks, including facial emotion recognition, sentiment analysis, and spoken command detection. To quantify shifts on classification APIs, We adopt the following metrics:

- **Prediction Overlap**. Prediction overlap measures how often an ML API's prediction on the same input remains the same at different evaluation periods. Formally, it can be expressed as

$$PO(t_1, t_2) \triangleq \frac{1}{|D|} \sum_{(x,y) \in D} \mathbb{1} \{f(x, t_1) = f(x, t_2)\}$$

  Here, $t_1$ and $t_2$ are two evaluation time periods. $PO = 1$ indicates an ML API's predictions do not change, and $PO = 0$ means its predictions between $t_1$ and $t_2$ are completely different.

- **Confidence Movement.** API shifts include both prediction and confidence score changes. For a fixed data point, an ML API's prediction can remain the same, but its confidence may still move up and down over time. To measure this, we use confidence movement

$$CM(t_1, t_2) \triangleq \frac{\sum_{(x,y) \in D} \mathbb{1} \{f(x, t_1) = f(x, t_2)\} \cdot [q(x, t_1) - q(x, t_2)]}{\sum_{(x,y) \in D} \mathbb{1} \{f(x, t_1) = f(x, t_2)\}}$$

  If $CM(t_1, t_2) > 0$, then among all data points without prediction shifts, the evaluated ML API is more confident at time $t_1$ than at time $t_2$. If $CM(t_1, t_2) < 0$, then on average, the API's confidence is less confident at time $t_1$ than at time $t_2$. It is worth noting that many applications are sensitive to confidence changes. For example, a customer review application may trust an ML API's prediction if its confidence is larger than a threshold, and involve a human expert otherwise. Even if all predictions stay the same, the confidence change over time may still mitigate or worsen the human expert's workload.

- **Model Accuracy.** One of the most widely adopted ML API assessments is accuracy, i.e., how often the ML API makes the right prediction. Given a dataset $D$ and the label prediction $f(\cdot, t)$ by an ML API evaluated at time $t$, accuracy is simply

$$a(t) \triangleq \frac{1}{|D|} \sum_{(x,y) \in D} \mathbb{1} \{f(x, t) = y\}$$

  Thus, it is natural to quantify how the accuracy of an ML API changes over time.

- **Group Disparity.** Various metrics [24, 39, 58] have been proposed to quantify ML fairness. In this paper, we adopt one common metric called *group disparity* [39]. Suppose the dataset $D$ is partitioned into $K$ groups $D_1, D_2, \cdots, D_K$ by some sensitive feature (e.g., gender or race). Then group disparity is

$$GD(t) \triangleq \max_i \frac{1}{|D_i|} \left( \sum_{(x,y) \in D_i} \mathbb{1} \{f(x, t) = y\} \right) - \min_i \frac{1}{|D_i|} \left( \sum_{(x,y) \in D_i} \mathbb{1} \{f(x, t) = y\} \right)$$

  In a nutshell, group disparity measures the accuracy difference between the most privileged group and the most disadvantaged group. Larger group disparity implies more unfairness, and $GD(t) = 0$ implies the API achieves perfect fairness at time $t$.

**A case study on DIGIT.** We start with a case study on a spoken command recognition dataset, DIGIT [4]. DIGIT contains 2,000 short utterances corresponding to digits from 0 to 9, and the task is to predict which number each utterance indicates. We have evaluated three speech recognition APIs from IBM, Google, and Microsoft in year 2020, 2021, and 2022, separately. The utterances were spoken by people with US accent, French accent, and German accent. Thus, we use accent as the sensitive feature to group the data instances and then measure the group disparity.

As shown in Figure 1, there are many interesting observations in this case study. First, the accuracy changes are substantial: for example, as shown in Figure 1(a), Google API's accuracy increased by 20% from 2020 to 2021. The prediction changes are even more significant: For example, from 2020 to 2021, IBM API's accuracy rose by 4% (see Figure 1(a), but 1-79.4%=20.6% predictions by IBM API were changed (see Figure 1(b)). This is perhaps because while some mistakes were fixed by the API update, some utterances previously correctly predicted may be predicted incorrectly by the updated version. Even when the predictions remain steady, the confidence score can still significantly move up or down. For example, the confidence produced by Microsoft API moved up by 31.7% from 2020 to 2021 (as shown in Figure 1(b)). Yet, its accuracy dropped by 1.5% (as shown in Figure

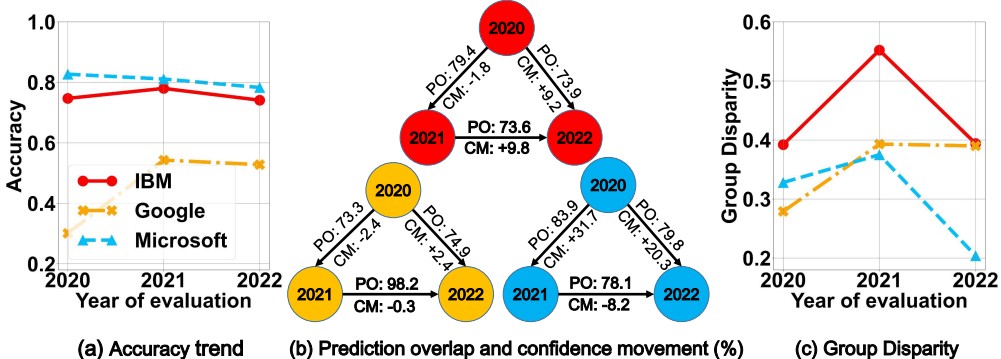

Figure 1: A case study on the dataset DIGIT. (a): accuracy over time. (b): prediction overlap and confidence movement of IBM, Google, and Microsoft APIs. (c): group disparity with respect to speaker accent. Overall, accuracy changes due to API shifts are notable, but the prediction changes are even more significant. For example, from 2020 to 2021, the accuracy of IBM API has increased by 4% (see (a)), but 20.6% predictions are actually different (see (b)). In addition, the confidence can move up by up to 31.7% (Microsoft from 2020 to 2021 in (a)) while the prediction accuracy slightly drops (Microsoft in (b)). This calls for cautions in confidence-sensitive applications. It is also worth noting that large group disparity exists for all evaluated APIs. Interestingly, API update over time may either improve or hurt overall accuracy as well as group fairness.

1(a)). This raises cautions in downstream applications that rely on confidence scores. It is also worth noting that group disparity exists for all evaluated APIs. In 2020, perhaps surprisingly, Google API's accuracy is the lowest but its disparity is also the smallest. Model update may either improve or hurt accuracy and group disparity. For example, Google API's accuracy is improved over time but its bias towards non-native accents is also worsen.

**Diverse API shifts across multiple classification tasks.**    Next we study API shifts across different tasks and datasets. For each API dataset pair, we calculate the prediction overlap and confidence movement between each evaluation time pair (2020–2021, 2020–2022, 2021–2022) and then report the results averaged over all time pairs. We also measure and compare its accuracy for each year. The results are shown in Figure 2.

Several interesting findings exist. First, small accuracy changes may be the result of large prediction shifts, i.e., small prediction overlaps. For example, about 10% predictions made by Amazon sentiment analysis API on IMDB (as shown in Figure 2(b1)) have changed, but its accuracy only changes by about 1% (Figure 2(a1)). Similarly, a 3% prediction difference exists for Microsoft API on RAFDB (Figure 2(b3)) while there is almost no change in its accuracy (Figure 2(a3)). This indicates general phenomena in API shifts: many API updates fix certain errors but also make additional mistakes. Next, we note that the API shifts are diverse. For spoken command recognition, all evaluated APIs' predictions are changed significantly (Figure 2(b1)). However, the shifts in APIs for facial emotion recognition is almost negligible (Figure 2(b3)). This implies that different APIs may be updated in a different rate and thus detecting whether a shift may have happened is useful. Moreover, different APIs' confidence movements are not similar. Sometimes an ML API tends to be more and more conservative: for example, the average confidences of Google API for spoken command recognition have dropped notably for all evaluated datasets. Sometimes an ML API becomes more and more confident: for example, Microsoft API for spoken recognition has increased its confidence over time on three out of four datasets ( Figure 2(c1)). More interestingly, its confidence may also depend on a dataset's property: as shown in Figure 2(c2), Amazon sentiment analysis API tends to be less confident on Chinese texts (WAIMAI and SHOP) but more confident on English texts (IMDB and YELP). Understanding how the confidence moves may help decision making in confidence-sensitive applications. We provide additional group disparity analysis in the appendix.

| | ML API | | IBM | | | Google | | | Microsoft | |
|---|---|---|---|---|---|---|---|---|---|---|---|
| Speech Comd | Year | 2020 | 2021 | 2022 | 2020 | 2021 | 2022 | 2020 | 2021 | 2022 |
| | DIGIT | 74.7 | 78.0 | 74.1 | 30.1 | 54.3 | 52.8 | 82.7 | 81.1 | 78.3 |
| | AMNIST | 98.3 | 91.2 | 98.5 | 88.5 | 96.0 | 95.7 | 98.6 | 98.8 | 98.5 |
| | CMD | 80.6 | 80.6 | 90.9 | 87.4 | 92.3 | 92.3 | 94.6 | 94.6 | 94.1 |
| | FLUENT | 88.8 | 88.9 | 91.9 | 96.9 | 97.5 | 97.5 | 97.5 | 97.9 | 98.1 |

(a1)

| ML API | IBM | Google | MS |
|---|---|---|---|
| DIGIT | 75.6 | 82.1 | 80.6 |
| AMNIST | 93.5 | 94.9 | 98.4 |
| CMD | 92.0 | 96.4 | 96.3 |
| FLUENT | 90.7 | 99.1 | 98.2 |

(b1)

| IBM | Google | MS |
|---|---|---|
| 5.7 | -1.7 | 14.6 |
| -0.7 | -7.3 | 7.6 |
| 0.8 | -4.9 | 5.5 |
| 0.2 | -3.8 | -1.1 |

(c1)

| | ML API | | Amazon | | | Google | | | Baidu | |
|---|---|---|---|---|---|---|---|---|---|---|---|
| Sentiment Anal | Year | 2020 | 2021 | 2022 | 2020 | 2021 | 2022 | 2020 | 2021 | 2022 |
| | IMDB | 79.1 | 78.0 | 78.1 | 86.4 | 86.4 | 86.4 | 51.6 | 51.6 | 51.6 |
| | YELP | 86.9 | 88.9 | 88.9 | 95.7 | 95.7 | 95.7 | 51.4 | 51.4 | 51.4 |
| | WAIMAI | 82.4 | 84.9 | 84.9 | 81.9 | 81.9 | 81.9 | 89.0 | 89.0 | 89.0 |
| | SHOP | 89.0 | 90.5 | 90.5 | 87.8 | 87.8 | 87.8 | 92.2 | 92.2 | 92.2 |

(a2)

| ML API | AMZ | Google | Baidu |
|---|---|---|---|
| IMDB | 88.6 | 100.0 | 100.0 |
| YELP | 95.7 | 100.0 | 100.0 |
| WAIMAI | 94.2 | 100.0 | 100.0 |
| SHOP | 95.9 | 100.0 | 100.0 |

(b2)

| AMZ | Google | Baidu |
|---|---|---|
| -7.8 | 0.0 | 0.0 |
| -3.2 | -0.1 | 0.0 |
| 1.6 | 0.0 | 0.0 |
| 0.3 | 0.0 | 0.0 |

(c2)

| | ML API | | Microsoft | | | Google | | | Face++ | |
|---|---|---|---|---|---|---|---|---|---|---|---|
| Facial Emotion | Year | 2020 | 2021 | 2022 | 2020 | 2021 | 2022 | 2020 | 2021 | 2022 |
| | FER+ | 81.4 | 84.4 | 84.4 | 67.7 | 67.7 | 67.7 | 68.4 | 68.4 | 68.4 |
| | RAFDB | 71.7 | 71.7 | 71.7 | 65.6 | 65.7 | 65.7 | 61.2 | 61.8 | 61.2 |
| | EXPW | 72.8 | 72.8 | 72.8 | 66.3 | 65.2 | 65.2 | 62.2 | 62.2 | 62.2 |
| | AFNET | 72.3 | 72.3 | 72.3 | 68.3 | 68.3 | 68.3 | 64.1 | 64.1 | 64.1 |

(a3)

| ML API | MS | Google | Face++ |
|---|---|---|---|
| IMDB | 97.4 | 99.9 | 100.0 |
| YELP | 97.4 | 99.9 | 100.0 |
| WAIMAI | 99.6 | 93.9 | 100.0 |
| SHOP | 99.6 | 99.8 | 100.0 |

(b3)

| MS | Google | Face++ |
|---|---|---|
| -0.6 | 0.0 | 0.0 |
| -0.1 | -0.1 | 0.0 |
| -0.4 | -1.4 | 0.0 |
| -0.1 | -0.1 | 0.0 |

(c3)

Figure 2: Summary on classification API shifts from 2020 to 2022. Tables on row 1, 2 and 3 correspond to spoken command recognition, sentiment analysis, and facial emotion recognition, respectively. (a1)-(a3): Accuracy of each year. (b1)-(b3): Average prediction overlap. (c1)-(c3): Average confidence movement. Units: %. Red and green indicate low and high values, respectively. The accuracy changes exhibit various patterns overall, while the API shifts are also diverse: all spoken command recognition APIs' predictions have been changed significantly during the past years, while significant changes exist for only one third of the APIs for the other two tasks. Confidence movements are also interesting. For example, Google API for spoken command recognition tends to be less confident (c1), while Amazon sentiment API is more confident on Chinese texts but less on English texts (c2).

## 4.2 Findings on Structured Prediction APIs

Next we turn to the structured prediction APIs. Similar to standard classification APIs, we use prediction overlap to measure prediction changes due to shifts of structured predictions APIs. For each data instance, we use the average of all predicted labels' confidences as an overall confidence, and then still apply confidence movement to quantify how an API's confidence shifts over time. To measure structured prediction API's performance, we adopt the standard multi-label accuracy

$$ma(t) \triangleq \frac{1}{|D|} \sum_{(x,y) \in D} \frac{|f(x,t) \cap y|}{|f(x,t) \cup y|}$$

Finally, we keep using group disparity to evaluate fairness of an ML API, but replace the 0-1 loss $\mathbb{1}\{f(x,t) = y\}$ by the Jaccard similarity $\frac{|f(x,t) \cap y|}{|f(x,t) \cup y|}$.

**A case study on COCO.** We start with a case study on the dataset COCO. COCO contains more than a hundred thousand images, and the goal is to determine if one or more objects from 80 categories show up in each image. We have evaluated three APIs from Microsoft, Google, and EPixel, respectively. To measure group disparity, we adopt the gender labels [73] for a subset of COCO which contains a person, and then calculate the group disparity on this subset for all evaluated ML APIs. The results are summarized in Figure 3.

Our first observation is that the accuracy shift can be quite large, leading to an "accuracy cross". As shown in Figure 3(a), EPixel API's accuracy drops by more than 20% while Google API's accuracy increases by 1%. Consequentially, Google API becomes more accurate than EPixel, while the latter was more accurate in 2020. This implies that API shifts can be impactful in business decision making such as picking which ML API to use. In addition, the prediction shifts are much larger than those for simple classification APIs. For example, prediction overlap for EPixel is less than 30%, meaning

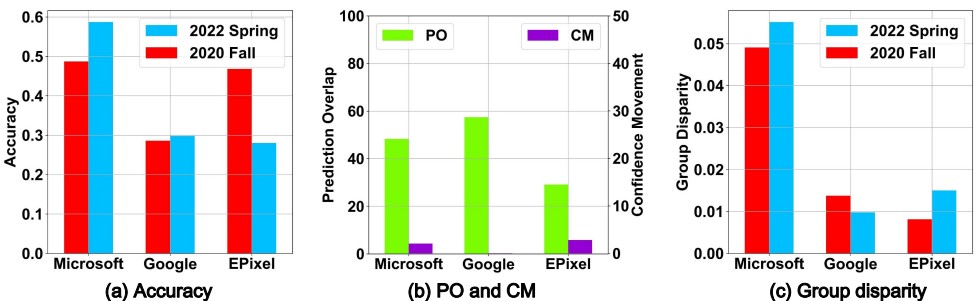

Figure 3: A case study on the dataset COCO. (a): accuracy over time. (b): prediction overlap (%) and confidence movement (%). (c): group disparity with respect to gender. Here, the accuracy change is quite significant. E.g., EPixel API update leads to 20% accuracy drop (as shown in (a)). Prediction shifts are also large: prediction overlap can be less than 30% (as shown in (b)). The confidence movement is relatively small. It is also worth noting that high accuracy does not imply better fairness. In fact, Microsoft API's accuracy is the highest, but its group disparity is also the largest (c).

that 70% of the predictions are different than before (as shown in Figure 3(b)). On the other hand, the confidence movement is relatively small: as shown in Figure 3(b), no API's confidence movement is larger than 3%. It is also worth noting that high accuracy does not imply better fairness necessarily. In fact, Microsoft API's accuracy is the highest, but its group disparity is also the largest. As shown in Figure 3(a) and (c), API shifts may improve the accuracy but simultaneously amplify the group disparity: Microsoft API's accuracy increases by 10% but its group disparity is also enlarged.

**Various API shift patterns across structured prediction tasks.** Finally, we dive deeply into various API shift patterns for more structured prediction tasks. The prediction overlaps, confidence movements, and accuracy changes for 27 API-dataset pairs are summarized in Figure 4.

There are several interesting observations. First, the accuracy changes are significant for multi-label image classification but relatively small for the other two tasks, as shown in Figure 4(a1)-(a3). However, API shifts for structured predictions are more common than classification tasks. In fact, as shown in Figure 4(b1)-(b3), prediction changes occur for almost all ML APIs. The magnitudes of the shifts are also larger. This is perhaps because structured prediction is more sensitive to model updates than those for classifications. The confidence movement is relatively small though. Note that confidence movements do not always reflect the APIs' performance changes. For instance, EPixel API's confidence increases on all evaluated datasets, but its accuracy actually drops. This is probably because EPixel's update removes a label due to low confidence but this label was part of the true label set. Detecting, estimating, and explaining such phenomena is needed for robustly adopting ML APIs.

# 5 Additional Discussions and Maintenance Plans

**More frequent evaluations.** ML APIs are increasingly growing and updated frequently. Thus, we plan to enrich our database by continuously evaluating ML APIs more frequently, i.e., every 6 months. As of 2022 August, we have collected additional predictions of all structured prediction APIs. As shown in Figure 5, significant prediction changes already occurred in 6 months. For example, the accuracy of IBM named entity API on the GMB dataset dropped from 50% (March 2022) to 45% (August 2022). Those newly collections have been added to our database. More details can be found in the appendix.

**Comparison with open-source ML models.** As a baseline, we have also measured the performance of several open source ML models on all datasets. As shown in Table 4, the open source ML models' performance varies across different datasets, and can be sometimes better than that of commercial APIs. This further emphasizes the importance of monitoring commercial APIs' performance.

**Expansion of Datasets and ML APIs.** Part of the future plan is to expand the scope of datasets and ML APIs. To allow this, we plan to solicit needs from the ML communities: a poll panel will be created on our website, and ML researchers, engineers, and domain experts are all welcome to vote

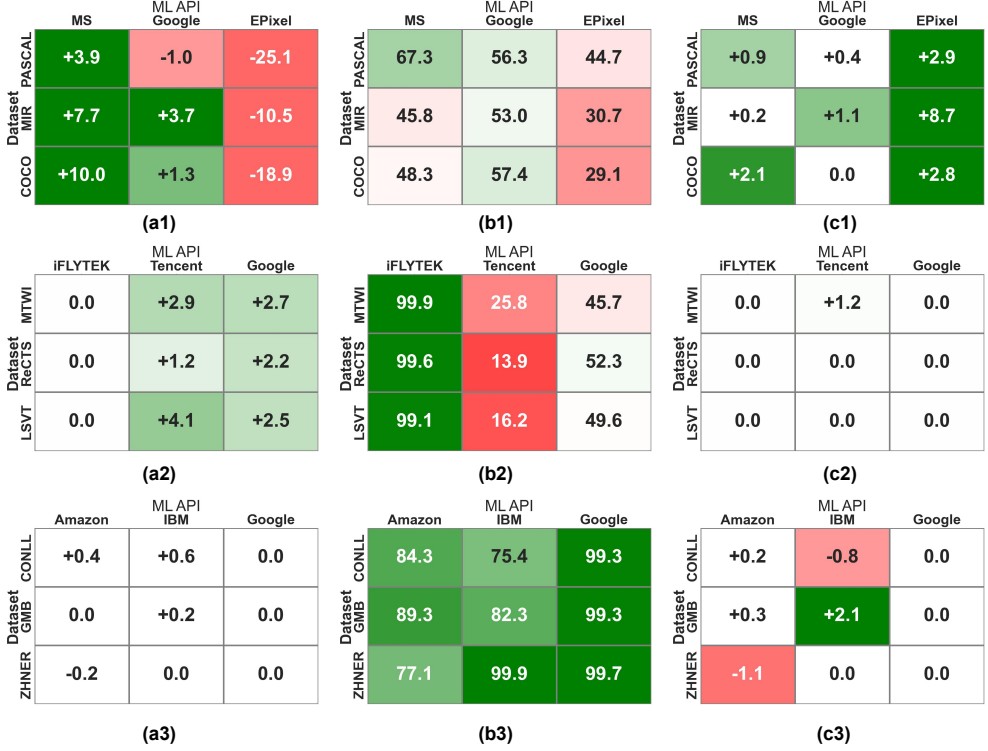

Figure 4: Summary on structured prediction API shifts. Row 1, 2, and 3 correspond to multi-label image classification, scene text recognition and named entity recognition. The left, middle, and right column correspond to accuracy changes (%), prediction overlaps (%), and confidence movements (%), respectively, between 2020 and 2022. The accuracy change is large for multi-label image classification but relatively small for the other two tasks. Predicted labels change notably for many of the evaluated ML APIs. The confidence movement is relatively small, though.

for which ML APIs and which datasets to include in our database. We will periodically update the database based on the community's feedback.

# 6 Conclusions and Open Questions

ML APIs play an increasingly important role in real world ML adoptions and applications, but there are only a limited number of papers studying the properties and dynamics of these commercial APIs. In this paper, we introduce HAPI, a large scale dataset consisting of samples from various tasks annotated by ML APIs over multiple years. Our analysis on HAPI shows interesting findings, including large price gaps among APIs for the same task, prevalent ML API shifts between 2020 and 2022, diverse performance differences between API venders, and consistent subgroup performance disparity. And this is just scratching the surface. HAPI enables many interesting questions to be studied in the ML marketplaces. A few examples include:

- How to determine which API or combination of APIs to use for any given application? HAPI can serve as a testbed to evaluate and compare different API calling strategies.
- How to perform unsupervised or semi-supervised performance estimation under ML API shifts? This is useful for practical ML monitoring but not possible without a detailed ML API benchmark over time provided by HAPI.
- Generally, how to estimate performance shifts when both ML APIs and data distributions shift?
- How to explain the performance gap due to ML API shifts? More fine-grained understanding of how the API's prediction behavior changes over time would be useful for practitioners.

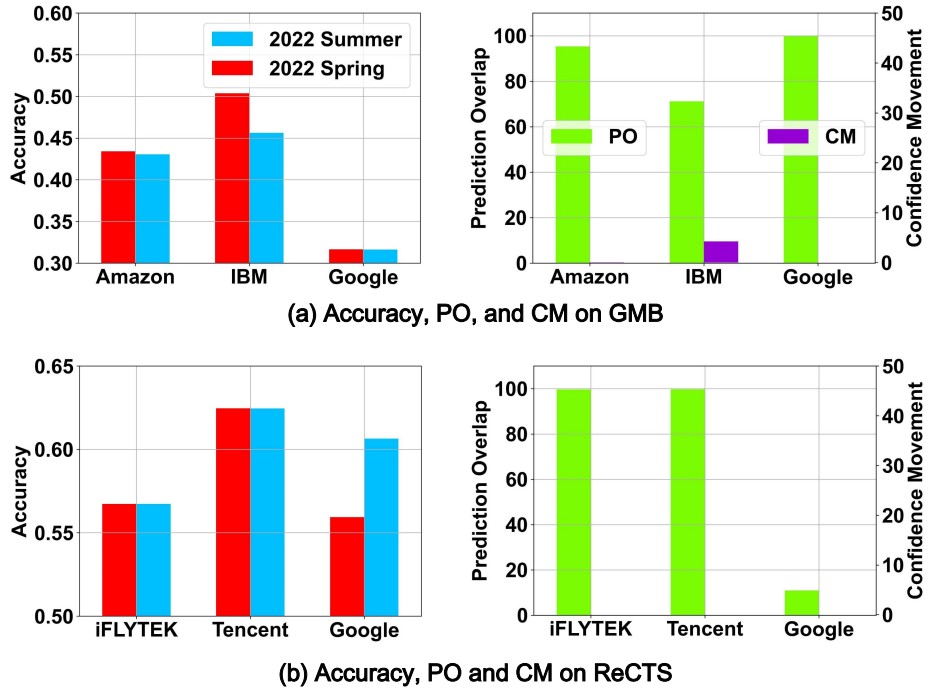

Figure 5: API Shifts within 6 months. (a) and (b) correspond to the GMB and ReCTS datasets, respectively. Overall, significant prediction and accuracy occurred in 3 out 6 ML APIs.

Table 4: Performance of open source models on the evaluated datasets. For some tasks, open source models' performance can be even better than that of the commercial APIs.

| Task | Speech Recognition | | | | Multi-label Image Classification | | |
|---|---|---|---|---|---|---|---|
| Open source model | DeepSpeech [23] | | | | SSD [18] | | |
| Dataset | DIGIT | AMNIST | CMD | FLUENT | PASCAL | MIR | COCO |
| Performance | 0.60 | 0.92 | 0.80 | 0.87 | 0.64 | 0.25 | 0.40 |
| Task | Sentiment Analysis | | | | Scene Text Recognition | | |
| Open source model | Vader [52] | | | | PP-OCR [43] | | |
| Dataset | IMDB | YELP | WAIMAI | SHOP | MTWI | ReCTS | LSVT |
| Performance | 0.69 | 0.75 | 0.64 | 0.78 | 0.63 | 0.51 | 0.47 |
| Task | Facial Emotion Recognition | | | | Named Entity Recognition | | |
| Open source model | A convolution neural network [15] | | | | Spacy [17] | | |
| Dataset | FER+ | RAFDB | EXPW | AFNET | CONLL | GMB | ZHNER |
| Performance | 0.77 | 0.60 | 0.56 | 0.64 | 0.53 | 0.55 | 0.63 |

These and other open questions enabled by HAPI are increasingly critical with the growth of ML-as-a-service. HAPI can greatly stimulate more research on ML marketplace. All of the data in HAPI is openly available at `http://hapi.stanford.edu/`.

## Acknowledgement

This project is supported in part by NSF CCF 1763191, NSF CAREER AWARD 1651570 and NSF CAREER AWARD 1942926. There is no industrial funding. We appreciate the anonymous reviewers for their invaluable feedback.

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
