# Supplementary materials

The supplementary materials include additional details of HAPI and extra model shift study.

## A  Additional Details of HAPI

In this section, we provide additional details of the constructed dataset HAPI, including motivation, composition, collection process, preprocessing and cleaning, uses, distribution, and maintenance.

### A.1  Motivation

HAPI was created to enable research on ML APIs. This includes but is not limited to, for example, determing which API or combination of APIs to use for different user data or applications as well as budget constraints, estimating how much performance has changed due to API shifts, and explaining the performance gap due to ML API shifts.

### A.2  Composition, and Collection Process

Each instance in HAPI consists of a query input for an API (e.g., an image or text) along with the API's output prediction/annotation and confidence scores. For example, one instance could be an image from the image dataset COCO [57], and {*(person, 0.9), (sports ball, 0.78), (tennis racket, 0.45)*}, the associated annotation by Microsoft API. This means Microsoft API predicts three labels, *person, sports*, and *tennis racket*, with confidence scores *0.9, 0.78*, and *0.45*, respectively.

The query inputs were collected from 21 datasets for 6 different tasks. For SCR, four datasets were used: DIGIT [4], AMNIST [27], CMD [70], and FLUENT [59]. The sampling rate is 8 kHz, 48 kHz, 16 kHz, and 16 kHz, respectively. Each utterance is a spoken digit (i.e., 0-9) in DIGIT and AMNIST and a short command from a total of 30 commands (such as "go", "left", "right", "up", and "down") or white noises in CMD. In FLUENT, the commands are typically a phrase (e.g., "turn on the light" or "turn down the music") from a total of 248 phrases. Four text datasets were used for SA: YELP [21], IMDB [60], SHOP [16], and WAIMAI [20]. YELP and IMDB are both in English while WAIMAI and SHOP are in Chinese. FER+ [25], RAFDB [56], EXPW [72], and AFNET [61] were used for FER task. The emotion labels were anger, disgusting, fear, happy, sad, surprise, and natural.

Three datasets were used for MIC: PASCAL [44], MIR [51], and COCO [57]. There are 20 and 80 distinct labels in PASCAL and COCO, respectively. MIR contains 25 unique labels, and we removed the label "night" as it is not in the label set of any ML APIs. For STR, we adopted MTWI [48], ReCTS [71], and LSVT [67], three datasets containing real world images with Chinese texts. Finally, for NER task, we used CONLL [65], ZHNER [22], and GMB [28]. CONLL and GMB contain both English texts while ZHNER is a Chinese text dataset.

For each instance in those datasets, we have evaluated the prediction from the mainstream ML APIs from 2020 to 2022. HAPI was collected from 2020 to 2022. For classification tasks, the predictions/annotations of each API were collected in the spring of 2020, 2021, and 2022. For structured predictions, all APIs' predictions were collected in fall 2020 and spring 2022, separately. The details can be found in Table 1.

### A.3  Preprocessing and Cleaning

This includes both (i) preprocessing on the original inputs to the ML APIs and (ii) cleaning of the ML APIs' outputs. The preprocessing on the original inputs is as follows. On FLUENT, all 248 unique phrases were mapped to 31 unique commands as provided in the original source [59]. The original labels in YELP are user ratings (1,2,3,4, and 5). 1 and 2 were transformed to negative; 3, 4, and 5 were mapped to positive. IMDB, WAIMAI and SHOP contain polarized review labels and thus we directly used those labels. As a result, classification on all SA datasets is a binary task. We used a sampled version of YELP: 10,000 text paragraphs with label positive and negative separately were randomly drawn from the original YELP dataset. The original IMDB dataset has been partitioned into training and testing splits, and thus we used its testing split, including 25,000 text paragraphs. All instances in WAIMAI and SHOP were used. The facial images in FER+ was the same as the FER dataset from the ICML 2013 Workshop on Challenges in Representation. A training and testing

split and regenerated labels are provided in FER+. We adopted the testing split with the regenerated labels. RAFDB and AFNET contain images for both basic emotions (anger, fear, disgusting, happy, sad, surprise, and natural) and compound emotions. We only evaluated ML APIs on images for basic emotions, as all evaluated ML APIs focus on basic emotions. Different from FER+, RAFDB, and AFNET, an image in EXPW may contain multiple faces. Thus, the labels include both the bounding box and the labelling workers' confidence. Thus, we extracted aligned faces as ML APIs' inputs by enlarging by 10% and then cropping the provided face bounding boxes whose confidence scores are larger than 0.6.

Less preprocessing was performed for structured prediction datasets. For MIC, we directly sent all raw images to the ML APIs. A diverse collection of images is included for STR: images for advertising sales forms MTWI, while most images in ReCTS are photos taken on sing boards. LSVT's iamges are typically street view images. While all images in MTWI and ReCTS are fully annotated, LSVT contains both fully and partially annotated images. HAPI only considers the images with full annotations as inputs to ML APIs. For NER datasets, all samples were included in HAPI. Yet, we only focused on three widely used types of entities: person, location, and organization.

Different ML APIs may use different label sets for the same tasks. For example, both "disgust" and "disgusting" may be returned by different ML APIs to refer to the same facial emotion. Thus, label alignment is needed. For classification tasks, we manually matched each API's predicted labels to a unique number. For example, for FER datasets, both "happy" and "happiness" were mapped to label 3, and label 4 corresponded to "sad", "sadness", and "unhappiness". For MIC with less than 100 unique labels, we were able to create the label maps manually too. On STR datasets, predictions (i) that are within 0-9 or (ii) whose unicode is in the range of u4e00-u9fff are maintained. For NER, we also manually mapped each API's entity type to a universal type. For example, "people" and "human" are both mapped to "person".

### A.4   Uses, Distribution, and Maintenance

HAPI has been tested and used in this paper at the time of publication. It can be used in any research related to ML prediction APIs or marketplaces, too. We will also maintain an incomplete list of which papers or projects have been developed on top of HAPI.

The dataset is publicly available on the internet. The dataset is distributed on Lingjiao Chen's website: `https://github.com/lchen001/HAPI` under Apache License 2.0. It was first released in 2022. The dataset will be maintained by Lingjiao Chen and other authors of this paper. In addition, HAPI will also be updated every few months to include up-to-date predictions from the mainstream ML APIs as well as emerging ML APIs. All the details and updates can be accessed on `https://github.com/lchen001/HAPI`.

## B   Extra Model Shift Study

We provide accuracy and group disparity study on two more datasets, AMNIST and RAFDB. On AMNIST, we again use the accents of different speakers to group the datasets. The accents in AMNIST cover "German", "South Korean", "Spanish", "Madras", "Levant", "English", "Chinese", "Brasilian", "Italian", "Egyptian American", "South African", "Arabic", "Danish", "French" and "Tamil". On RAFDB, we use race ("Caucasian", "African-American", and "Asian") to partition the dataset. The results are shown in Figure 6.

Several interesting observation exist. Overall, there are a large accuracy differences and group disparity for both datasets. For instance, as shown in Figure 6(b), the group disparity of IBM can vary from 0.05 to 0.20. Noting the accuracy drop of IBM API during the same time period (2020-2021) is relatively smaller, one might infer that IBM's accuracy drop is due to worse ability to recognize certain accents. On the other hand, Microsoft API's overall accuracy on AMNIST seems to be stable (less than 0.3% as shown in Figure 6(a)), but there is a significant change in its group disparity (larger than 3% as shown in Figure 6(a)). On RAFDB, the change over time is relatively smaller (Figure 6(c) and (d)). Yet, APIs with better accuracy exhibits lower group disparity. For example, Face++ API's accuracy is the lowest, and its group disparity is also the higest. Thus, it still remains an interesting question to relate accuracy and group disparity changes due to API shifts. How to determine which

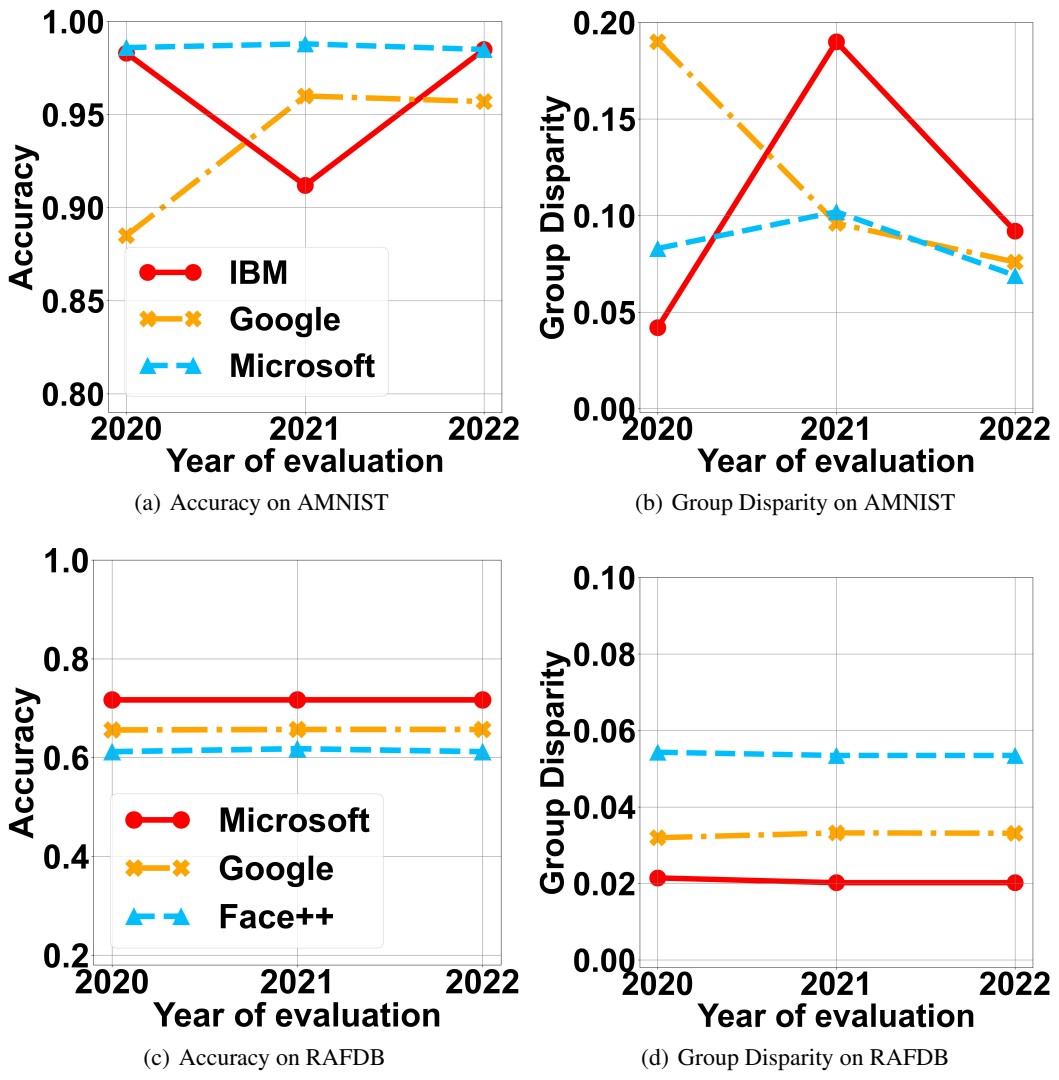

(a) Accuracy on AMNIST

(b) Group Disparity on AMNIST

(c) Accuracy on RAFDB

(d) Group Disparity on RAFDB

Figure 6: Additional accuracy and group disparity study. The two rows correspond to AMNIST and RAFDB respectively. Overall, there are large accuracy differences and group disparity for both cases. The group disparity on AMNIST is much larger than that on RAFDB, although thie former's accuracy is also higher. This further verifies that higher overall accuracy does not necessarily lead to better fairness.

API or combination of APIs to use for different user data, budget constraints, accuracy and fairness targets is also enabled by HAPI and open to the community.

## C  Additional Discussions

**Potential overfitting of commercial APIs on the publicly available datasets.**  We suspect that the commercial APIs do not overfit the datasets we used for evaluations for three reasons. First, the terms of use for many of the datasets disallow commercial applications. For example, the RAFDB dataset is "available for non-commercial research purposes only" (see the webpage `http://www.whdeng.cn/RAF/model1.html`). Second, the performance of most evaluated APIs is well below that of typical overfitting, which is often more than 90%. Third, we observed that several APIs' performances dropped over time. For example, the EPixel API's accuracy on the COCO dataset

dropped from 47% (Fall 2020) to 27% (Spring 2022), as shown in Figure 3 (a). This shows that it is still very interesting to compare commercial APIs over time on these datasets.

**Licenses and restrictions enforced by the ML APIs.**   The terms of use for most ML APIs (see, e.g., `https://cloud.google.com/terms` and `https://azure.microsoft.com/en-us/support/legal/`) require no sublicensing to a third party. However, to the best of our knowledge, they do not prevent evaluating and analyzing those APIs' performance. In fact, evaluating and comparing the performance of different cloud services is not only desired by users but also encouraged by cloud providers. For example, Google Cloud provides its own performance measurement tool (`https://cloud.google.com/free/docs/measure-compare-performance#:˜:text=Google%20Cloud%20Platform%20provides%20two,%2Dto%2Ddate%20and%20unbiased`). This is probably because a systematic study of the ML APIs can help the providers improve their services. For example, gender shade [30], the seminal work on bias and stereotypes embedded in face detection APIs, has helped ML API providers improve their services and thus been appreciated by the industry. We hope HAPI enables better understanding of the commercial ML APIs and in turn helps API providers build better services too.

**Maintenance and development plans for HAPI.**   The maintenance and development plans consist of three main parts. First, we will continuously evaluate all ML APIs considered in the paper. Currently the evaluation is planned to occur every 6 months. If significant performance changes are consistently observed every 6 months, the update frequency will be further increased, say, to every 3 months or every month. Second, we plan to enlarge the set of ML APIs, datasets, and tasks in HAPI. MLaaS is an increasingly growing industry, and new ML APIs are launched from time to time. Thus, we plan to add the evaluation of the emerging ML APIs every 6 months. It is also important to include more representative and diverse datasets and document how quality of the datasets affects ML APIs' performance. For example, for image classification, ML APIs' robustness to the image resolution and natural noises (such as rain and snow) can largely influence practitioners' choices. Last but not least, the usefulness of a database is determined by our community. Thus, we plan to implement an interactive feedback system on our website to collect opinions from our community. This helps, for example, solicit preference of which datasets, ML APIs, and tasks to include in HAPI.

As a first step, we have collected the predictions from ML APIs for all structured tasks, including multi-label image classification, scene text recognition, and named entity recognition, in August 2022. The accuracy changes and prediction overlap as well as confidence movement compared to the prediction collected 6 months ago (February or March 2022) are shown in Figure 7 and Figure 8, respectively. Overall, we observe accuracy shifts of several APIs. For example, the accuracy of the IBM API for named entity recognition on the GMB dataset dropped from 50% (March 2022) to 45% (August 2022), as shown in Figure 7 (e). The performance of the Google scene text recognition API was 60% in the ReCTS dataset in August 2022, which was 4% higher than that in March 2022 as shown in Figure 7 (h). In fact, prediction changes of the Google API occurred on more than 80% of images in ReCTS as well as MTWI and ReCTS, as shown in Figure 8 (g), (h) and (i). There was little prediction changes of multi-label image classification APIs. The confidence scores of the IBM API increased on CONLL (by 4%) and GMB (by8%) and remained almost the same for the other APIs, as shown in Figure 8 (d) and (e). Overall, this analysis suggests that significant changes can happen within six months and thus frequent updates of the database is needed.

**Choices of Datasets and ML APIs.**   We chose existing datasets for a few reasons. First, the ML community is familiar with the datasets and they are relatively well annotated and evaluated. Second, those datasets can be easily assessed on the internet. Third, those datasets covered a diverse range of real-world scenarios (for example, the COCO dataset included objects in outdoor/indoor environments, at a small/large scale, and with different brightness). In fact, based on our conversation with many practitioners, there is a large interest in understanding commercial APIs' performance on those datasets. Thus it is a good starting point to evaluate ML APIs on those popular datasets.

Similarly, the selection criteria for ML APIs are (i) popularity, (ii) easy access for users, and (iii) representation of diverse companies. Based on our discussion with practitioners, Google APIs are widely used and easily accessible and hence included in our database. ML APIs from domain-specific companies such as EPixel, Face++, and iFLYTEK were also included to increase the representativeness of our database.

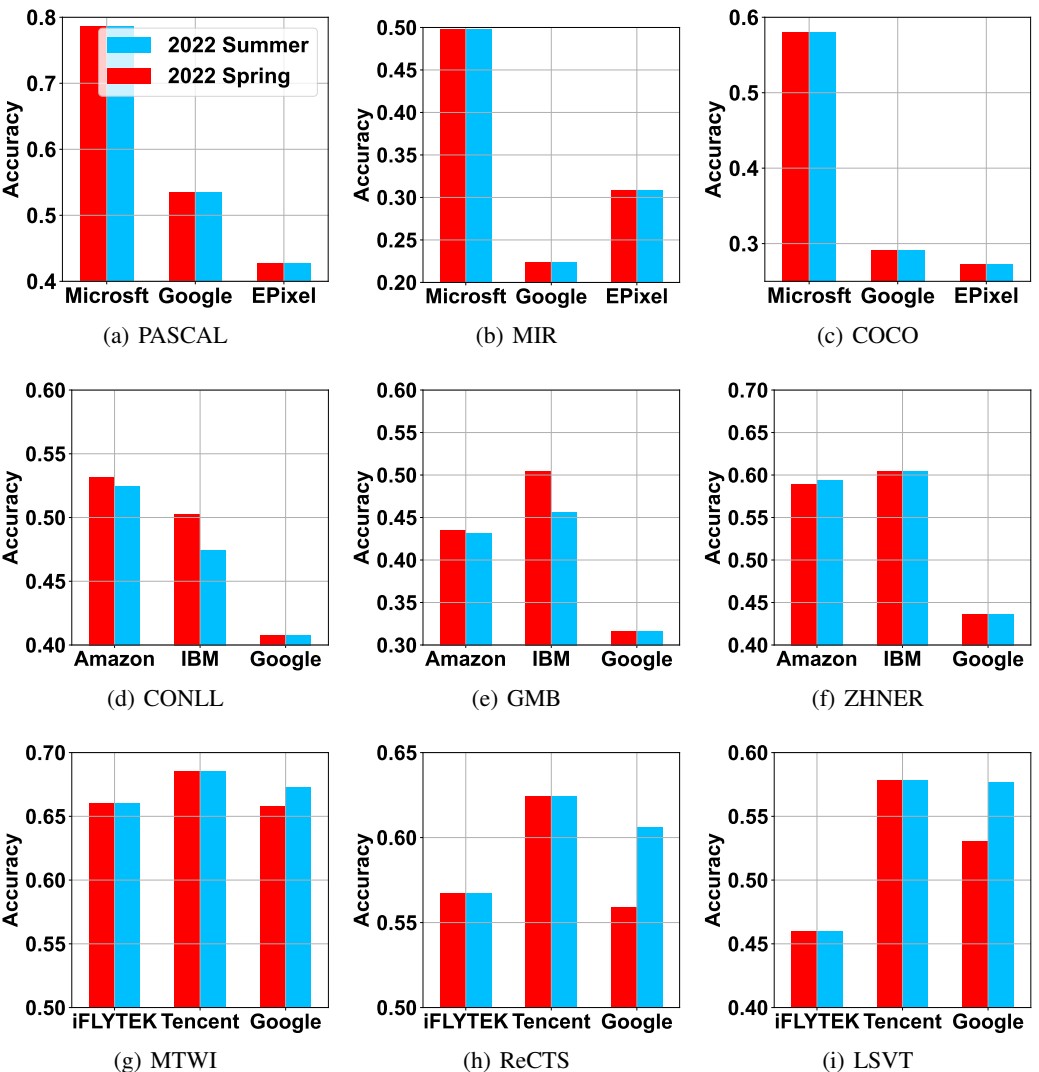

Figure 7: Accuracy changes of structured prediction APIs within 6 months (2022 Spring – 2022 Summer). The first, second, and third row corresponds to multi-label image classifications, scene text recognition, and named entity recognition. Overall, we observe accuracy shifts of several APIs. For example, IBM named entity API's performance dropped on CONLL and GMB, while the accuracy of Google scene text API increased on MTWI and ReCTS.

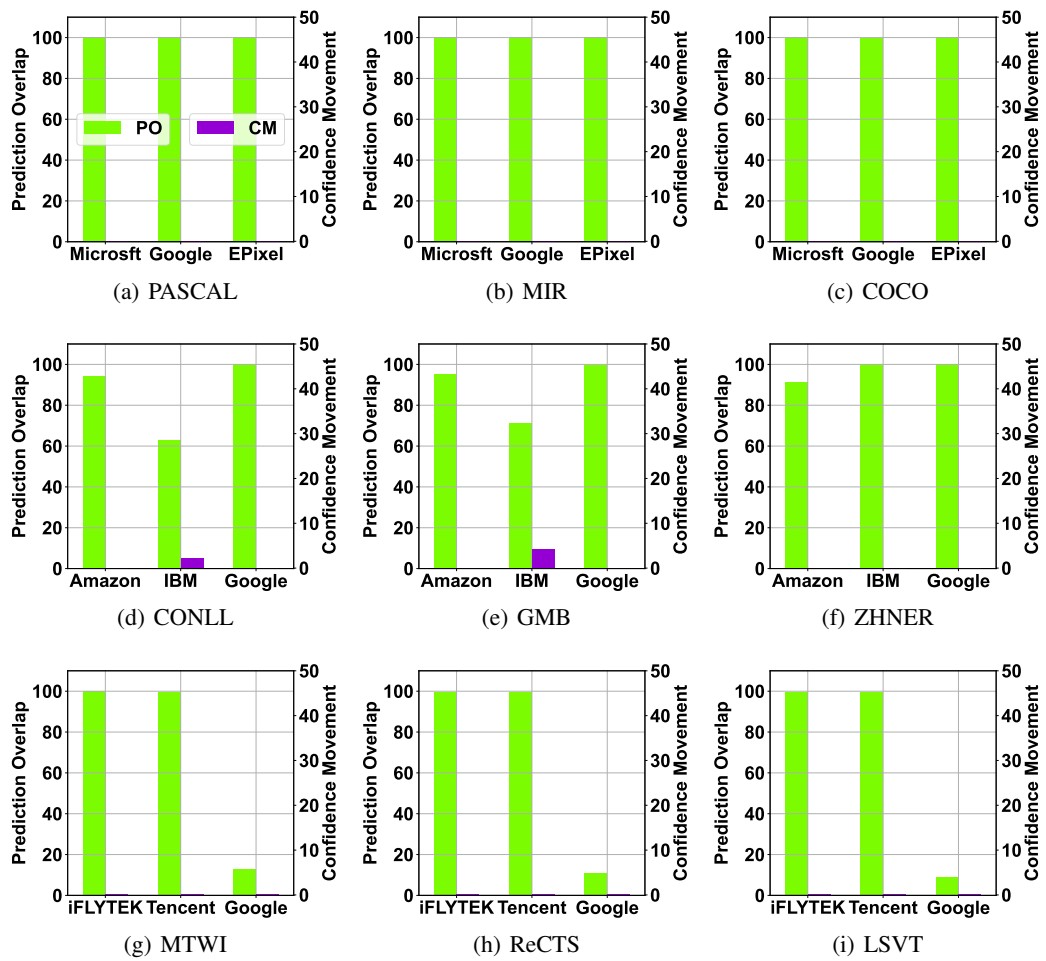

Figure 8: Prediction overlap and confidence movement of structured prediction APIs within 6 months (2022 Spring – 2022 Summer). The first, second, and third row correspond to multi-label image classifications, scene text recognition, and named entity recognition. Overall, most prediction shifts occurred for scene text and named entity recognition. There was little prediction change of multi-label image classification APIs. The confidence scores of the IBM API increased on CONLL and GMB and remained almost the same for the other APIs.

**Responsible usage of facial emotion datasets.** According to the original documents of the facial emotion datasets (FER+ [25], RAFDB [56], EXPW [72], and AFNET [61]), all the face images in these four facial emotion datasets were collected via querying search engines (e.g., Google, Bing, and Yahoo!) with certain keywords (e.g., happy faces). While the images are publicly retrievable from search engines, we did not find clear documentation of the individual consent process for these datasets. We recognize that facial photos are sensitive data, and will remove photos from HAPI upon request. Moreover, photos curated online may not fully represent the general public, and emotion annotations can be subjective and noisy. Therefore, analysis of these datasets should be interpreted with care. For example, the fact that an API's performance on some of these datasets changes over time is important to know, while the absolute performance across different datasets may not be directly comparable. We will continue to work with the machine learning community to expand HAPI to include high-quality benchmark datasets.

**Additional outputs from ML APIs.** Several APIs generate information beyond confidence scores and predicted labels. For instance, for multi-label image classification, Microsoft vision API provides the bounding boxes for all detected objects. Given a text paragraph, Google sentiment analysis API returns not only a predicted attitude label with a confidence score, but also a magnitude score

indicating how significant the detected attitude is. HAPI allows users to query the raw outputs including the above information, too.

**Relations to model stealing attacks and defenses.** Model stealing attacks [62, 69] and defenses [53, 63] have raised large attentions in both security and ML communities. HAPI provides a large set of predictions from real-world ML APIs to study model stealing attacks and defenses. An interesting next step, for example, is to benchmark different model stealing attacks on HAPI. It is also interesting to study if applying model inversion attacks [45] on the stolen model can steal the training datasets of commercial ML APIs.

**Strength and weakness of the evaluated datasets.** Recall that our dataset selection criteria are (i) popularity, (ii) easy access, and (iii) diversity. Now we provide more details about how the selected datasets meet the criteria and what limitations remain. We start by the speech command recognition datasets. They are all widely studied by the speech command recognition community (for example, CMD [70] has been cited more than 700 times since published in 2018) and are easily accessible on the internet. They contain a diverse range of commands: DIGIT and AMNIST mainly obtain spoken digits, while CMD and FLUENT contain more complicated commands such as "turn on the light in the kitchen". Their varying sampling rates also cover different application scenarios. In addition, speaker information is also provided, enabling fairness study. A potential limitation is that all those datasets are clean, i.e., there is almost no environmental noise in the utterances. Evaluating ML APIs' robustness to such noise is an interesting next step. Similarly, the datasets for SA and FER are also widely used. For example, the IMDB [60] dataset has been cited more than 3,000 times, and the citation of RAFDB [56] is above 700. Besides easy access on the internet, they also represent diverse data distribution: YELP, IMDB, WAIMAI cover user reviews for restaurants, movies, and delivery services, while feedback for items from various category is included in SHOP. Limitations include (i) that only English and Chinese texts are included, and (ii) that most text paragraphs are short. FER+ contains gray and low-resolution images, while RAFDB, EXPW and AFNET consist of colored images with high resolutions. One limitation is that only few images contain more than one person.

Similarly, the datasets used for the structured prediction tasks are also widely used and easily accessible. For example, PASCAL [44] and COCO [57] are perhaps the most widely studied datasets for object recognition. MTWI [48], ReCTS [71], and LSVT [67] are one of the largest scene text recogntion datasets and were used for competitions in International Conference on Document Analysis and Recognition (ICDAR), one of the flagship conferences on document analysis. CONLL [65] and GMB [28] are widely studied for named entity recognition in English, while the GitHub repository hoding the Chinese named entity recongition dataset, ZHNER [22], has received almost two thousand stars. Besides easy access, they also cover different scenarios. For example, most images are low-resolution in PASCAL but high-resolution in COCO and MIR. MTWI contains mostly advertising images, while most images are photos taken on sign boards in ReCTS and on street view in LSVT. A natural way to extend the diversity of the datasets is to evaluate vision APIs on images with large number of labels (e.g., larger than 1000). It is also interesting to study how ML APIs perform on multilingual scene text images. Multilingual text datasets with domain specifications are also useful to understand named entity recognition APIs. Continuously identifying and evaluating ML APIs on more diverse datasets is part of our future plans.

**Support of AI ethics.** HAPI enables the study of AI ethics on a range of commercial systems targeting various tasks. For example, predictions of vision APIs on human objects can be used to study the biases and stereotypes on sensitive features including races, genders, and ages. The evaluation of speech APIs opens the door for understanding and comparing how accents and nationality of the speakers affect different ML APIs' performance. Besides understanding the real-world APIs' ethic issues, how to efficiently detect and estimate those issues can also be explored on top of HAPI. For example, one may use the heterogeneity of the predicted labels between different population groups to detect an API's biases. In addition, HAPI offers an opportunity to explore whether and how the biases and stereotypes can be mitigated by adaptively selecting which API to use. In a nutshell, HAPI supports various studies of trustworthy AI on a range of commercial APIs.

# D  Datasheet

This section includes a "datasheet" for the dataset, following the outline proposed by [46].

### D.1 Motivation

**For what purpose was the dataset created?** HAPI was created to enable research on ML APIs. This includes but is not limited to, for example, determining which API or combination of APIs to use for different user data or applications as well as budget constraints, estimating how much performance has changed due to API shifts, and explaining the performance gap due to ML API shifts.

**Who created the dataset?** The dataset was created in the Zou Group at Stanford University.

**Who funded the creation of the dataset?** This project is supported in part by NSF CCF 1763191, NSF CAREER AWARD 1651570 and NSF CAREER AWARD 1942926.

### D.2 Composition

**What do the instances that comprise the dataset represent? What data does each instance consist of?** Each instance in HAPI consists of a query input for an API (e.g., an image or text) along with the API's output prediction/annotation and confidence scores. For example, one instance could be an image from the image dataset COCO [57], and {*(person, 0.9), (sports ball, 0.78), (tennis racket, 0.45)*}, the associated annotation by Microsoft API. This means Microsoft API predicts three labels, *person, sports*, and *tennis racket*, with confidence scores *0.9, 0.78*, and *0.45*, respectively.

**How many instances are there in total (of each type, if appropriate)?** As of 08/2022, There are a total of 1,761,417 instances in the dataset. For a breakdown by task and dataset, see Table 3

**Is any information missing from individual instances?** Not that the authors are aware of.

**Are relationships between individual instances made explicit (e.g., users' movie ratings, social network links)?** In some of the datasets, *e.g.* FER [26] relationships between instances are provided where applicable.

**Are there recommended data splits (e.g., training, development/validation, testing)?** There are no recommended data splits.

**Are there any errors, sources of noise, or redundancies in the dataset?** There are no errors or sources of noise known to the authors.

**Is the dataset self-contained, or does it link to or otherwise rely on external resources (e.g., websites, tweets, other datasets)?** The dataset is not self-contained and relies on a number of previously released datasets. For a comprehensive list of these datasets,

**Does the dataset contain data that might be considered confidential?** HAPI is based on existing, external datasets. It does not introduce any new data that may be considered confidential, but the authors cannot speak to the confidentiality of the external datasets.

**Does the dataset contain data that, if viewed directly, might be offensive, insulting, threatening, or might otherwise cause anxiety?** HAPI includes predictions from MLaaS APIs. These APIs may demonstrate societal bias that could be viewed as offensive or insulting. However, the authors are not aware of any such instances in the dataset. Additionally, HAPI is based on existing, external datasets. The authors of HAPI are unaware of offensive content in these external datasets. MLaaS APIs

**Does the dataset identify any subpopulations (e.g., by age, gender)?** The dataset does not include annotations for any subpopulations.

**Is it possible to identify individuals (i.e., one or more natural persons), either directly or indirectly (i.e., in combination with other data) from the dataset?** HAPI is based on existing, external datasets. It does not introduce any new data that could aid in the identification of individuals, but the external datasets may include data in which it is possible to identify individuals.

**Does the dataset contain data that might be considered sensitive in any way?** See questions above.

### D.3 Collection Process

**How was the data associated with each instance acquired?** For each instance in those datasets, we have evaluated the prediction from the mainstream ML APIs from 2020 to 2022. HAPI was collected

from 2020 to 2022. For classification tasks, the predictions/annotations of each API were collected in the spring of 2020, 2021, and 2022. For structured predictions, all APIs' predictions were collected in fall 2020 and spring 2022, separately. The details can be found in Table 1.

**What mechanisms or procedures were used to collect the data (e.g., hardware apparatuses or sensors, manual human curation, software programs, software APIs)?** We used software APIs for MLaaS providers to collect the data.

**If the dataset is a sample from a larger set, what was the sampling strategy (e.g., deterministic, probabilistic with specific sampling probabilities)?** HAPI relies on a number of external datasets. It includes the full set of instances from these external datasets. The sampling strategy for each external dataset is not known to the authors of HAPI. The external datasets were chosen based on a few different criteria. First, the ML community is familiar with the datasets and they are relatively well annotated and evaluated. Second, those datasets can be easily assessed on the internet. Third, those datasets covered a diverse range of real-world scenarios (for example, the COCO dataset included objects in outdoor/indoor environments, at a small/large scale, and with different brightness).

**Who was involved in the data collection process (e.g., students, crowdworkers, contractors) and how were they compensated (e.g., how much were crowdworkers paid)?** The data collection process was performed by the authors of HAPI.

**Over what timeframe was the data collected?** The MLaaS predictions were collected between 2020 and 2022. We will continue to collect predictions every six months going forward.

**Were any ethical review processes conducted (e.g., by an institutional review board)?** No ethical review processes were conducted.

**Did you collect the data from the individuals in question directly, or obtain it via third parties or other sources (e.g., websites)?** HAPI is based on existing, external datasets which may include data collected from individuals.

**Were the individuals in question notified about the data collection?** The authors of HAPI are unaware of the notification policies used by the external datasets on which HAPI is based.

**Did the individuals in question consent to the collection and use of their data?** The authors of HAPI are unaware of the consent policies used by the external datasets on which HAPI is based.

**Has an analysis of the potential impact of the dataset and its use on data subjects (e.g., a data protection impact analysis) been conducted?** The authors of HAPI have not conducted any analysis of the potential impact of the dataset and its use on data subjects.

### D.4   Pre-processing/cleaning/labeling

**Was any preprocessing/cleaning/labeling of the data done** The preprocessing on the original inputs is as follows. On FLUENT, all 248 unique phrases were mapped to 31 unique commands as provided in the original source [59]. The original labels in YELP are user ratings (1,2,3,4, and 5). 1 and 2 were transformed to negative; 3, 4, and 5 were mapped to positive. IMDB, WAIMAI and SHOP contain polarized review labels and thus we directly used those labels. As a result, classification on all SA datasets is a binary task. We used a sampled version of YELP: 10,000 text paragraphs with label positive and negative separately were randomly drawn from the original YELP dataset. The original IMDB dataset has been partitioned into training and testing splits, and thus we used its testing split, including 25,000 text paragraphs. All instances in WAIMAI and SHOP were used. The facial images in FER+ was the same as the FER dataset from the ICML 2013 Workshop on Challenges in Representation. A training and testing split and regenerated labels are provided in FER+. We adopted the testing split with the regenerated labels. RAFDB and AFNET contain images for both basic emotions (anger, fear, disgusting, happy, sad, surprise, and natural) and compound emotions. We only evaluated ML APIs on images for basic emotions, as all evaluated ML APIs focus on basic emotions. Different from FER+, RAFDB, and AFNET, an image in EXPW may contain multiple faces. Thus, the labels include both the bounding box and the labelling workers' confidence. Thus, we extracted aligned faces as ML APIs' inputs by enlarging by 10% and then cropping the provided face bounding boxes whose confidence scores are larger than 0.6.

Less preprocessing was performed for structured prediction datasets. For MIC, we directly sent all raw images to the ML APIs. A diverse collection of images is included for STR: images for

advertising sales forms MTWI, while most images in ReCTS are photos taken on sing boards. LSVT's iamges are typically street view images. While all images in MTWI and ReCTS are fully annotated, LSVT contains both fully and partially annotated images. HAPI only considers the images with full annotations as inputs to ML APIs. For NER datasets, all samples were included in HAPI. Yet, we only focused on three widely used types of entities: person, location, and organization.

Different ML APIs may use different label sets for the same tasks. For example, both "disgust" and "disgusting" may be returned by different ML APIs to refer to the same facial emotion. Thus, label alignment is needed. For classification tasks, we manually matched each API's predicted labels to a unique number. For example, for FER datasets, both "happy" and "happiness" were mapped to label 3, and label 4 corresponded to "sad", "sadness", and "unhappiness". For MIC with less than 100 unique labels, we were able to create the label maps manually too. On STR datasets, predictions (i) that are within 0-9 or (ii) whose unicode is in the range of u4e00-u9fff are maintained. For NER, we also manually mapped each API's entity type to a universal type. For example, "people" and "human" are both mapped to "person".

**Was the "raw" data saved in addition to the preprocessed/cleaned/labeled data (e.g., to support unanticipated future uses)?** The raw unprocessed predictions are included and can be accessed via our Python API.

**Is the software that was used to preprocess/clean/label the data available?** The software for preprocessing the data is not currently available but will be provided soon.

### D.5   Uses

**Has the dataset been used for any tasks already?** HAPI has been tested and used in this paper at the time of publication. It can be used in any research related to ML prediction APIs or marketplaces, too. We will also maintain an incomplete list of which papers or projects have been developed on top of HAPI.

**Is there a repository that links to any or all papers or systems that use the dataset?** As the authors become aware of papers or systems that use HAPI, we will maintain a list of them on the project website `https://github.com/lchen001/HAPI`.

**What (other) tasks could the dataset be used for?** See Section 6 for a list of potential tasks.

**Is there anything about the composition of the dataset or the way it was collected and pre-processed/cleaned/labeled that might impact future uses?** The dataset relies on a limited set of existing datasets – in the future, we plan to expand the set of datasets that we use to include more diverse and up-to-date datasets.

**Are there tasks for which the dataset should not be used?** The authors of HAPI do not know of any particular tasks for which using this dataset should be avoided.

### D.6   Distribution

**Will the dataset be distributed to third parties outside of the entity (e.g., company, institution, organization) on behalf of which the dataset was created?** Yes.

**How will the dataset will be distributed (e.g., tarball on website, API, GitHub)?** The dataset is publicly available on the internet.

**When will the dataset be distributed?** The dataset is publicly available on the internet. Instructions for downloading the dataset and using the Python API are available at `https://github.com/lchen001/HAPI`.

**Will the dataset be distributed under a copyright or other intellectual property (IP) license, and/or under applicable terms of use (ToU)?** The dataset is distributed under the Apache License 2.0.

**Have any third parties imposed IP-based or other restrictions on the data associated with the instances?** The authors of HAPI are not aware of any third parties imposing IP-based or other restrictions on the data associated with the instances.

**Do any export controls or other regulatory restrictions apply to the dataset or to individual instances?** The authors of HAPI are not aware of any export controls or other regulatory restrictions applying to the dataset or to individual instances.

### D.7    Maintenance

**Who will be supporting/hosting/maintaining the dataset?** The authors of HAPI will be supporting/hosting/maintaining the dataset.

**How can the owner/curator/manager of the dataset be contacted (e.g., email address)?** Reach out to Lingjiao Chen (lingjiao [at] stanford [dot] edu) and Sabri Eyuboglu (eyuboglu [at] stanford [dot] edu).

**Is there an erratum?** There is currently no erratum.

**Will the dataset be updated (e.g., to correct labeling errors, add new instances, delete instances)?** First, we will continuously evaluate all ML APIs considered in the paper. Currently, the evaluation is planned to occur every 6 months. If significant performance changes are consistently observed every 6 months, the update frequency will be further increased, say, to every 3 months or every month. MLaaS is an increasingly growing industry, and new ML APIs are launched from time to time. Thus, we plan to enlarge the set of ML APIs, datasets, and tasks in HAPI as well.

**If the dataset relates to people, are there applicable limits on the retention of the data associated with the instances (e.g., were the individuals in question told that their data would be retained for a fixed period of time and then deleted)?** HAPI is based on publicly available datasets. The retention policies of these datasets vary.

**Will older versions of the dataset continue to be supported/hosted/maintained?** Yes.

**If others want to extend/augment/build on/contribute to the dataset, is there a mechanism for them to do so?** Yes, potential contributors are encouraged to contact the authors of HAPI or submit a pull request on GitHub.