# OpenReview forum: "HAPI: A Large-scale Longitudinal Dataset of Commercial ML API Predictions"
_NeurIPS.cc/2022/Track/Datasets_and_Benchmarks — NeurIPS 2022 Datasets and Benchmarks _

### Official Review · Reviewer_V17d · 2022-07-26
**Review of Chen & Jin & Eyuboglu & Re & Zaharia & Zou (2022) - updated 20220819**

**Rating:** 7
**Confidence:** 3

**Strengths:**

Strengths:
1. Significance - the contribution is highly significant given today's trends in using MLAAS for cost-effective processing of large collections of data.
2. Relevance - The ML community can gain awareness about issues (from accuracy to ethics) about the 'black box' MLAASes by observing their [emergent] behaviour over time - and such a dataset as HAPI will be beneficial in tracking these changes.
3. Accessibility and Accountability - The compilation of such a dataset is documented in Section A, specifically e.g. A.4, of the Supplementary Material. It is based on existing state-of-the art datasets - see [Ethics] and [Relation To Prior Work] below.
4. Ethical and Social Implications - The HAPI dataset might help in research on e.g. fairness and bias of MLAASes by uncovering the general 'shift' in trends and emergent behaviour over time. See [Ethics] below as well.

**Weaknesses:**

Weaknesses:
1. Significance - there is no issue regarding significance of this contribution.
2. Relevance - there is a potential issue with the choice of the MLAASes surveyed in the dataset, which might lead to arguments against using the dataset. e.g. Google APIs are overrepresented in the dataset (across the "6 areas" as alluded to in the [Summary And Contributions]). See detailed commentary in [Correctness] below.
3. Accessibility and Accountability - per (2), there is an overrepresentation of Google APIs in the study, with an underrepresentation of Amazon APIs. See [Correctness].
4. Ethical and Social Implications - closely related to (2) and (3) - to convince readers of its impartiality, the selection criteria needs to be made clear - why the particular distribution of cloud providers? This has impacts in terms of, say, social implications - if a ML practitioner needs to compare the top MLAAS providers, and seeks to use HAPI, there will be missing data as the top 3 cloud providers are represented differently in the dataset.

**Additional Feedback:**

The main thing this reviewer is concerned about is the somewhat arbitrary (with apologies) choice of MLAAS per category without explaining why, as well as the limitations this could bring about - providing this explanation including Funding will alleviate any potential doubts on conflicts-of-interests etc.

**Clarity:**

The paper is generally well-written, albeit with some issues hampering readability.

1. There are typos through the paper. Examples include but are not limited to "Table 3: Datesets " (p.4), "Octomber 2020" (p. 3 in Table 2).

2. The tables/figure(s) are not clearly understandable at first glance.

(a) "Table 2: Prices of ML services used for each task at their evaluation time" - it is not clear what the triples mean, until the caption makes it clear (it is across X years)

(b) Table 3 - instead of using acronyms, there is enough space on the page to include a brief description of "SCR" etc (usage of acronyms are fine, but if the datasets themselves are also abbreviated, this will hamper readability).

(c) Figure 2 - which is a table but with conditional formatting - it is not clear how the table is laid-out at first glance as it is, in effect, a multidimensional table (A1 ... C3 containing 9 cells; each cell containing a mini-table).  The units could also be added to the table itself (% vs points).
Other issues about the layout: "Row 1, 2 and 3 represent spoken command recognition, sentiment analysis, and facial emotion recognition, respectively." - shouldn't this be the first set A1/A2/A3 (5 rows incl header), ... etc, for there are 5 rows to a "set"? Also the labels "spoken command recognition" etc could appear next to a band. There are no row labels for the final "horizontal set" (i.e. C1/C2/C3)



**Correctness:**

There are no issues in-the-main with regards to the assemblage of test datasets (chosen based on extant state-of-the-art), metrics adopted (e.g., S4.1), nor the standardisation of output instances (S3) as "the output formats of different ML APIs are often different".

However, the main issue which needs clarification, per [Weaknesses], point (3) -
This disparity of the surveyed MLAAS 'brands' are at odds with, e.g., popularity of Cloud MLAAS provide ratings (e.g., by survey - https://www.kdnuggets.com/2021/01/cloud-computing-data-science-ml-trends-2020-2022-battle-giants.html; or by market share - https://www.statista.com/chart/18819/worldwide-market-share-of-leading-cloud-infrastructure-service-providers/).

Hence, this raises an interesting point - what is the selection criteria for the surveyed MLAASes - cost? funding? In fact, the selection criteria for the MLAASes are not documented in the main nor supplementary materials manuscript. This needs to be made explicitly clear to avoid any potential conflict of interest - see also [Ethics] below. Moreover, the funding for this project (which involves monetary considerations in using the MLAASes) needs to be disclosed.

**Documentation:**

As per [Relation To Prior Work] - assumptions and issues related to its constituent datasets need to be documented.

As per [Correctness] - details on MLAAS selection needs to be made explicitly clear to dispel any potential conflict-of-interest.

Cannot seem to find a long-term maintenance plan beyond what is in Supplementary A.4.


**Ethics:**

S5 seems to have a brief discussion of "Conclusions and Open Questions" focusing mainly on accuracy and MLAAS selection. It would be good if the authors can include statements on how their work can support research in the realm of AI Ethics, including how their HAPI dataset can be used to systematically detect biases which are still open problems in MLAAS.

The reviewer notes that the authors have included some considerations for diversity in constituent datasets.

Funding/sponsorship is not disclosed.

**Relation To Prior Work:**

The authors have claimed "HAPI is the first large-scale ML API dataset", which the reviewer accepts *prima facie*.
However, the authors need to discuss, at least in the Supplementary Material, the background information on the hows/whats of the original datasets used in constructing HAPI.

e.g. in the Supplementary Material, "for SCR, four datasets were used: DIGIT [4], AMNIST [ 23], CMD [ 57 ], and FLUENT [ 49 ]. The sampling rate is 8 kHz, 48462 kHz, 16 kHz, and 16 kHz, respectively. " --> why these four? justifications including brief overviews about each sets strengths and weaknesses, including any foreseen limitations (see also [Documentation] below).

**Summary And Contributions:**

In summary, the submission is about HAPI, a longitudinal "large-scale dataset" of ML predictions by cloud ML providers (APIs, or MLAAS - ML As A Service). Specifically, it consists of ~1.7M instances (each instance consisting of input, API output and metadata incl. confidence scores)  across "diverse tasks including image tagging, speech recognition and text mining from 2020 to 2022" - specifically the 6 areas of "spoken command recognition (SCR), sentiment analysis (SA), and facial emotion recognition (FER), and structured predictions including multi-label image classification (MIC), scene text recognition (STR), and named entity recognition (NER)". This is potentially beneficial for the community to study the evolution/change of such 'black boxed' APIs -- from accuracy, to algorithms employed, to mitigation strategies towards ethical issues, etc.

- - -

### Revision 20220819
In light of the authors' willingness for an R&R to incorporate suggested changes from myself and other reviewers, and that a proportion of the key issues have been addressed, I have revised my rating accordingly. This note documents this change in the interest of transparency.
Score was `5` --> revised to `6`

### Revision 20220831
Considering the newest version of the paper, score was `6` --> revised to `7`

---

> ### Author Response · Authors · 2022-08-19
> **Thank you very much for your thoughtful review. We have updated the paper to incorporate your feedback. (1/2)**
>
> Thank you very much for your thoughtful review. We have carefully updated the paper to incorporate all of your feedback (please see uploaded files). The revision has strengthened the paper. We answer your questions as follows.
>
>
> ***[What are the selection criteria of MLaaS APIs?]***: The selection criteria for ML APIs are (i) popularity, (ii) easy access for users, and (iii) representation of diverse companies. Based on our discussion with practitioners, Google APIs are widely used and easily accessible and hence included in our database. ML APIs from domain-specific companies such as EPixel, Face++, and iFLYTEK were also included to increase the representativeness of our database. We have added a detailed discussion in the appendix (see line 661 - line 665, page 21).
>
>
>
> ***[What is the funding for this project?]***: We have added a statement on the funding sources in the appendix (see line 500 - line 502, page 15). This project received no industry funding and has no conflict of interest with any companies building the ML APIs.
>
> ***[Typos "Table 3: Datesets " (p.4), "Octomber 2020" (p. 3 in Table 2).]***: Thank you for pointing them out. We have fixed those typos.
>
> ***[Can you make the meaning of the triples in "Table 2” clearer?]***: To make the meaning clearer, We have broken the triples into separate columns and edited the column title accordingly.
>
> ***[Can you replace the acronyms of the tasks by brief descriptions in Table 3?]***: We have replaced task acronyms with brief descriptions (e.g., “SCR”-> “Speech Command Recog”) in Table 3.
>
> ***[Can you fix formatting issues with Figure 2?]***: We have edited Figure 2 accordingly. Specifically, we have added the unit info in the caption, replaced “Row 1, 2, and 3” with “Tables on row 1, 2, and 3”, and placed the task name in the corresponding Tables. The row labels for c1, c2, and c3 are the same for b1, b2, and b3, and thus omitted to save space.
>
> ***[Discuss the background information on the hows/whats of the original datasets used in constructing HAPI. Specifically, give justifications for choosing the four datasets for SCR.]***: The dataset selection criteria are (i) popularity in the ML community, (ii) easy access by users, and (iii) coverage of diverse subpopulations and applications. Take the four datasets used for speech command recognition as an example. They are all widely studied by the speech command recognition community (for example, CMD has been cited more than 700 times since published in 2018) and are easily accessible on the internet. They contain a diverse range of commands: DIGIT and AMNIST mainly include spoken digits, while CMD and FLUENT contain more complicated commands such as “turn on the light in the kitchen”. Their varying sampling rates also cover different application scenarios. In addition, speaker information is also provided, enabling fairness study. A potential limitation is that all those datasets are relatively clean, i.e., there is almost no environmental noise in the utterances. Evaluating ML APIs’ robustness to such noise is an interesting next step. We have added detailed discussions on the strengths and limitations of all datasets used in the appendix (see line 678 - line 710, page 22).
>
> ***[What is the long-term maintenance plan?]***: We have added detailed plans for updating and expanding HAPI going forward, consisting of three main parts. First, we will continuously evaluate all ML APIs considered in the paper. Currently, the evaluation is planned to occur every 6 months. If significant performance changes are consistently observed every 6 months, the update frequency will be further increased, say, to every 3 months or every month. Second, we plan to enlarge the set of ML APIs, datasets, and tasks in HAPI. MLaaS is an increasingly growing industry, and new ML APIs are launched from time to time. Thus, we plan to add the evaluation of the emerging ML APIs every 6 months. It is also important to include more representative and diverse datasets and document how the quality of the datasets affects ML APIs' performance. For example, for image classification, ML APIs' robustness to the image resolution and natural noises (such as rain and snow) can largely influence practitioners' choices. Last but not least, the usefulness of a database is determined by our community. Thus, we plan to implement an interactive feedback system on our website to collect opinions from our community. This helps, for example, solicit preference of which datasets, ML APIs, and tasks to include in HAPI. With that being said, we believe that the current HAPI dataset is already a valuable resource as it is the first large-scale evaluation of commercial ML APIs and contains data from more than two years. We have added a detailed discussion in the appendix (see line 627 - line 639, page 19).

---

> > ### Author Response · Authors · 2022-08-19
> > **Thank you very much for your thoughtful review. We have updated the paper to incorporate your feedback. (2/2)**
> >
> >
> > ***[It would be good if the authors can include statements on how their work can support research in the realm of AI Ethics.]***: Thank you for this helpful suggestion! One major strength of HAPI is that it enables the study of AI ethics on a range of commercial ML systems targeting various tasks. For example, predictions of vision APIs on human images can be used to quantify the biases and stereotypes on sensitive features including race, gender, and age. The evaluation of speech APIs opens the door for understanding and comparing how accents and nationalities of the speakers affect different ML APIs' performance. Besides understanding the real-world APIs' ethical issues, how to efficiently detect and estimate those issues can also be explored on top of HAPI. For example, one may use the heterogeneity of the predicted labels between different population groups to detect an API’s biases. In addition, HAPI offers an opportunity to explore whether and how the biases and stereotypes can be mitigated by adaptively selecting which API to use. In a nutshell,  HAPI supports various studies of trustworthy AI on a range of commercial APIs. We have added the detailed discussion in the appendix (see line 711 - line 720, page 22).
> >
> > Thank you again for your feedback, which has improved our paper! We hope you would consider increasing your score in light of our detailed response and revision. Thank you!

---

### Official Review · Reviewer_nKdr · 2022-07-27
**Interesting dataset + analysis; Likely to be a useful resource**

**Rating:** 8
**Confidence:** 4
**Clarity:** The paper is well written.

**Strengths:**

- The paper presents a valuable resource which would hard to gather for many other groups.
- The future applications which this dataset would allow seem important and currently relevant - eg. applications such as the ones related to MLaaS, ML pipeline monitoring (Sec 2), and the fairness centric analysis conducted in Sec 4.1.
- The coverage of tasks included in the dataset is good and the presented analysis is interesting.


**Weaknesses:**

- It seems important for the dataset to be updated over a longer period of time, its unclear what plan is in place to ensure that this happens.


**Additional Feedback:**

- Lines 115-122: These lines describe how all of the raw outputs are converted into a consistent format in this work. What additional information is present in the outputs from the different APIs? If there is substantial metadata it would be nice to release the additional output information in the hope that it will allow future work by others not envisioned in this paper.
- Is it possible to find/make educated guesses about the specific models (or kinds of models) or training datasets for the models whose outputs are included in this dataset? This might be useful additional information.
- In a project like this it would be meaningful to solicit datasets that the broader community would like to see predictions. If possible, it may be nice to have a system for community input on which datasets should be included in the dataset.
- Line 141: "PO=1" -> "PO=0"?
- Line 159: Consider using parenthesis to make the GD(t) equation easier to read.

**Post review responses:** Dear authors, thank you for the significant additions to the paper and your responses. I have raised my score appropriately.

**Correctness:**

The construction of the dataset seems sound.


**Documentation:**

I would highly encourage documenting this dataset in a datasheet (https://arxiv.org/pdf/1803.09010.pdf).


**Ethics:**

None.

**Relation To Prior Work:**

Prior work seems appropriately discussed.


**Summary And Contributions:**

The main proposal in the paper is a dataset of predictions of commercial machine learning apis. The tasks chosen are classification and structured prediction tasks with 3-4 datasets per task. The paper also promises to update this dataset of predictions from year to year. The paper also contains a analysis of the predictions in terms of overlap of predictions, changes in accuracy over time, changes in confidence and changes in performance by groups such as accents, race, and gender.

---

> ### Author Response · Authors · 2022-08-19
> **Thank you for your helpful feedback and support for the paper! (1/2)**
>
> Thank you for your helpful feedback and support for the paper! We answer your questions below and we have updated the paper to incorporate your suggestions.
>
> ***[Can HAPI be updated more frequently?]***: Yes. We plan to benchmark those APIs more frequently going forward, at least every 6 months, and continuously update HAPI. In fact, we just added evaluations in early August for all structured tasks. Specifically, these new collections include ML API predictions on all datasets for multi-label image classification (PASCAL, MIR, COCO), scene text recognition (MTWI, ReCTS, LSVT), and named entity recognition (CONLL, GMB, ZHNER). Compared to the evaluation in Spring 2022, non-negligible changes occurred in several APIs. For example, the accuracy of the IBM API for named entity recognition on the GMB dataset dropped from 50% (March 2022) to 45% (August 2022), as shown in Figure 5 (a). The performance of the Google scene text recognition API was 60% in the ReCTS dataset in August 2022, which was 4% higher than that in March 2022 as shown in Figure 5 (b). In fact, prediction changes of the Google API occurred on more than 80% of images in ReCTS. We have added a discussion to the main paper (see line 247- line 253 and Figure 5, page 9) and the appendix (see line 640 – line 653 and Figure 7 and Figure 8, page 19 - page 21).
>
> ***[What is the long-term plan?]***: We have added detailed plans for updating and expanding HAPI going forward, consisting of three main parts. First, we will continuously evaluate all ML APIs considered in the paper. Currently, the evaluation is planned to occur every 6 months. If significant performance changes are consistently observed every 6 months, the update frequency will be further increased, say, to every 3 months or every month. Second, we plan to enlarge the set of ML APIs, datasets, and tasks in HAPI. MLaaS is an increasingly growing industry, and new ML APIs are launched from time to time. Thus, we plan to add the evaluation of the emerging ML APIs every 6 months. It is also important to include more representative and diverse datasets and document how the quality of the datasets affects ML APIs' performance. For example, for image classification, ML APIs' robustness to the image resolution and natural noises (such as rain and snow) can largely influence practitioners' choices. Last but not least, the usefulness of a database is determined by our community. Thus, we plan to implement an interactive feedback system on our website to collect opinions from our community. This helps, for example, solicit preference of which datasets, ML APIs, and tasks to include in HAPI. With that being said, we believe that the current HAPI dataset is already a valuable resource as it is the first large-scale evaluation of commercial ML APIs and contains data from more than two years. We have added a detailed discussion in the appendix (see line 627 - line 639, page 19).
>
>
> ***[Can you document this dataset in a datasheet?]***: Thank you for the suggestion. We have added the datasheet in the appendix (see line 721 - line 894, page 23 - page 26).
>
>
> ***[What additional information is present in the outputs from the different APIs? It would be nice to release the additional output information.]***: Several APIs generate information beyond confidence scores and predicted labels. For instance, for multi-label image classification,  Microsoft vision API provides the bounding boxes for all detected objects. Given a text paragraph, Google sentiment analysis API returns not only a predicted attitude label with a confidence score but also a magnitude score indicating how significant the detected attitude is.
> We have added a discussion in the appendix (see line 666 - line 671, page 21 - page 22). The additional outputs are also available for download on our website.
>
> *** [Is it possible to find/make educated guesses about the specific models (or kinds of models) or training datasets for the models whose outputs are included in this dataset?]***: Model stealing attacks [61, 68] and defenses [52, 62] have raised large attention in both security and ML communities. HAPI enables the study of model stealing attacks and defenses on a large range of commercial ML APIs. An interesting next step, for example, is to benchmark different model stealing attacks on HAPI. It is also interesting to study if applying model inversion attacks [44] on the stolen model can steal the training datasets of commercial ML APIs. We have added a detailed discussion in the appendix (see line 672 - line 677, page 22).
>
> ***[Can you solicit datasets that the broader community would like to see predictions?]***: Thank you for this suggestion! Implementing an interactive feedback system on our website to collect opinions from our community has been added to the maintenance and development plan of HAPI. The detailed discussion can be found in the appendix (see line 627 - line 639, page 19).

---

> > ### Author Response · Authors · 2022-08-19
> > **Thank you for your helpful feedback and support for the paper! (2/2)**
> >
> > ***[Line 141: "PO=1" -> "PO=0"?]***: Thank you for pointing this typo out. We have fixed it.
> >
> > ***[Line 159: Consider using parentheses to make the GD(t) equation easier to read?]***: Thank you for this suggestion. We have added the parentheses in line 159.

---

### Official Review · Reviewer_82x1 · 2022-07-27
**Historical dataset of ML API queries and corresponding outputs from different providers**

**Rating:** 8
**Confidence:** 4

**Strengths:**

Accessibility: The dataset is easily accessible from the provided URL. A Python helper package is provided as well to make it easier to access slices of the dataset.

Significance: I agree with the authors that there’s a need for understanding the quality and shift of commercial ML APIs due to an increase in adoption of these products. The dataset contains 3 consecutive years of data, which is more than datasets of 1 or 2 years as referenced in related work.

Contribution: The authors consider different tasks and modalities when constructing the dataset. They also include example analyses regarding distribution shifts for different APIs.

**Weaknesses:**

Relevance: The datapoints are collected only once per year. Any analysis done using this dataset could be outdated by a year in the worst case.

**Additional Feedback:**

1. How were the dates for data collection chosen? Is it possible to increase the frequency of data collection?
2. Will the dataset continue to be updated in future years?

**Clarity:**

The paper is clear and it is easy to understand how the dataset was constructed.

**Correctness:**

The datasets used to query the commercial ML APIs are known to the research community.

**Documentation:**

The project website provides download links to the raw data, as well as instructions for using their Python package to query subsets of the dataset.

**Ethics:**

It is unclear to me whether the FER+, EXPW, RAFDB and AFNET facial emotion recognition datasets have been collected with consent of the human subjects.

**Relation To Prior Work:**

The authors compare to other datasets which only consider 1 or 2 consecutive years, as compared to this work where 3 consecutive years are considered. They also include downstream analyses in the related work.

**Summary And Contributions:**

The dataset is a collection of query, output pairs gathered at different points in time from commercially available ML APIs from different providers across a range of tasks. Besides the predictions, the dataset also includes the corresponding confidence scores as returned by each API.

The authors perform an example analysis regarding model shifts using the dataset collected.

---

### Revision Aug 19, 2022
The authors addressed most of the weaknesses pointed out by the reviewers. To my ethics concern however, I received a fallacious argumentum ad populum: "All of these are public benchmark datasets that are widely used in the machine learning community". I therefore lowered my score from 9 to 8.

---

> ### Author Response · Authors · 2022-08-19
> **Thank you for your helpful feedback and support for the paper!**
>
> Thank you for your helpful feedback and support for the paper! We answer your questions below and we have updated the paper to incorporate your suggestions.
>
> ***[Can HAPI be updated more frequently?]***: Yes. We plan to benchmark those APIs more frequently going forward, at least every 6 months, and continuously update HAPI. In fact, we just added evaluations in early August for all structured tasks. Specifically, these new collections include ML API predictions on all datasets for multi-label image classification (PASCAL, MIR, COCO), scene text recognition (MTWI, ReCTS, LSVT), and named entity recognition (CONLL, GMB, ZHNER). Compared to the evaluation in Spring 2022, non-negligible changes occurred in several APIs. For example, the accuracy of the IBM API for named entity recognition on the GMB dataset dropped from 50% (March 2022) to 45% (August 2022), as shown in Figure 5 (a). The performance of the Google scene text recognition API was 60% in the ReCTS dataset in August 2022, which was 4% higher than that in March 2022 as shown in Figure 5 (b). In fact, prediction changes of the Google API occurred on more than 80% of images in ReCTS. We have added a discussion to the main paper (see line 247- line 253 and Figure 5, page 9) and the appendix (see line 640 – line 653 and Figure 7 and Figure 8, page 19 - page 21).
>
>
>
> ***[Have the FER+, EXPW, RAFDB and AFNET facial emotion recognition datasets been collected with the consent of the human subjects?]***: Thank you for the question. All of these are public benchmark datasets that are widely used in the machine learning community. For example, the RAFDB paper has been cited more than 700 times in the past five years. We did not collect any of these image data ourselves. We wanted to use existing public datasets because they have been frequently used to benchmark algorithms (though not the commercially available ML APIs that we studied here).
>
>
> ***[Will the datasets be updated in the future?]***: Yes. We have added detailed plans for updating and expanding HAPI going forward, consisting of three main parts. First, we will continuously evaluate all ML APIs considered in the paper. Currently, the evaluation is planned to occur every 6 months. If significant performance changes are consistently observed every 6 months, the update frequency will be further increased, say, to every 3 months or every month. Second, we plan to enlarge the set of ML APIs, datasets, and tasks in HAPI. MLaaS is an increasingly growing industry, and new ML APIs are launched from time to time. Thus, we plan to add the evaluation of the emerging ML APIs every 6 months. It is also important to include more representative and diverse datasets and document how the quality of the datasets affects ML APIs' performance. For example, for image classification, ML APIs' robustness to the image resolution and natural noises (such as rain and snow) can largely influence practitioners' choices. Last but not least, the usefulness of a database is determined by our community. Thus, we plan to implement an interactive feedback system on our website to collect opinions from our community. This helps, for example, solicit preference of which datasets, ML APIs, and tasks to include in HAPI. With that being said, we believe that the current HAPI dataset is already a valuable resource as it is the first large-scale evaluation of commercial ML APIs and contains data from more than two years. We have added a detailed discussion in the appendix (see line 627 - line 639, page 19).

---

> > ### Comment · Reviewer_82x1 · 2022-08-19
> > **Revision Aug 19, 2022**
> >
> > The authors addressed most of the weaknesses pointed out by the reviewers. To my ethics concern however, I received a fallacious argumentum ad populum: "All of these are public benchmark datasets that are widely used in the machine learning community". I therefore lowered my score from 9 to 8.

---

### Official Review · Reviewer_AgWs · 2022-07-28
**Is it worth benchmarking a commercial ML API based on a standard dataset?**

**Rating:** 6
**Confidence:** 4
**Correctness:** why do we evaluate commercial APIs on…
**Clarity:** the paper is written well

**Strengths:**

- comparison of several commercial APIs could be used as a tool to choose the best API for our use case
- access to historical predictions of APIs

**Weaknesses:**

- the biggest concern: analysis of the commercial APIs based on standard datasets (a very high overfitting probability) proposes a limited use case for such results; it would be much better to propose a new dataset for such analysis; otherwise, it could be an analysis of which team added `this` dataset to the training data
- is a year a proper interval for the benchmark APIs?
- commercial APIs are compared, but there is a lack of some standard benchmarks; hence it is sometimes hard to compare with SOTA models

**Additional Feedback:**

1. Did you consider the comparison of the commercial APIs with open-source SOTA models? How does the difference look alike?
2. Why did you choose only standard datasets for the benchmark?
3. Do you plan to add more datasets?
4. How do you want to mitigate the potential overfitting of commercial models?

**Documentation:**

I would like to see more descriptions of how to benchmark commercial APIs with open source models - especially from the licensing point of view

**Ethics:**

- Lack of mention of when and how can we analyze commercial APIs and compare them. They have some restrictions on how they can be benchmarked in the licenses.

**Relation To Prior Work:**

it is hard to find previous related work

**Summary And Contributions:**

- analysis of how APIs change over time
- comparison of several commercial APIs over three years
- the opportunity to analyze the changes in ML models predictions

---

> ### Author Response · Authors · 2022-08-19
> **Thank you very much for your thoughtful review! We have updated the paper to incorporate your feedback. (1/2)**
>
> Thank you very much for your thoughtful review. We have carefully updated the paper to incorporate all of your feedback (please see uploaded files). We answer your questions as follows.
>
> ***[Do the commercial APIs overfit the standard datasets?]***: We believe that the commercial APIs do not overfit the datasets used for evaluations for three reasons. First, the terms of use for many of the datasets disallow commercial applications. For example, the RAFDB dataset is “available for non-commercial research purposes only” (see the webpage http://www.whdeng.cn/RAF/model1.html). Second, the performance of most evaluated APIs is well below that of typical overfitting, which is often more than 90%. Third, we observed that several APIs’ performance actually decreased over time. For example, the Everypixel API’s accuracy on the COCO dataset dropped from 47% (Fall 2020) to 27% (Spring 2022), as shown in Figure 3 (a). This shows that it is still very interesting to compare commercial APIs over time on these datasets. We have added the discussions in the appendix (see line 605 - line 613, page 19).
>
> ***[Can HAPI be updated more frequently?]***: Yes. We plan to benchmark those APIs more frequently going forward, at least every 6 months, and continuously update HAPI. In fact, we just added evaluations in early August for all structured tasks. Specifically, these new collections include ML API predictions on all datasets for multi-label image classification (PASCAL, MIR, COCO), scene text recognition (MTWI, ReCTS, LSVT), and named entity recognition (CONLL, GMB, ZHNER). Compared to the evaluation in Spring 2022, non-negligible changes occurred in several APIs. For example, the accuracy of the IBM API for named entity recognition on the GMB dataset dropped from 50% (March 2022) to 45% (August 2022), as shown in Figure 5 (a). The performance of the Google scene text recognition API was 60% in the ReCTS dataset in August 2022, which was 4% higher than that in March 2022 as shown in Figure 5 (b). In fact, prediction changes of the Google API occurred on more than 80% of images in ReCTS. We have added a discussion to the main paper (see line 247- line 253 and Figure 5, page 9) and the appendix (see line 640 – line 653 and Figure 7 and Figure 8, page 19 - page 21).
>
> ***[Did you consider the comparison of the commercial APIs with open-source SOTA models?]***: Yes, we have evaluated several open-source ML models’ performance on all datasets. Those open source models were chosen primarily due to their popularity and high performance. For example, the GitHub repositories for the PP-OCR and the Spacy library have received more than 24 thousand stars and 4 thousand forks, and are actively maintained by their developers and the communities. Overall, we observe that open-source models can sometimes outperform commercial APIs on specific datasets (e.g., on the AMNIST dataset, the DeepSpeech model gave a higher accuracy than Google API in 2020). We have added the results and discussions in the main paper (see line 254 – line 257 and Table 4, page 10).
>
> ***[What are the restrictions on when and how can we analyze commercial APIs and compare them due to those APIs’ licenses?]***: We discuss the dates of when we evaluated each API in Section 3 of the paper (line 101-103) and released our code for evaluating the APIs at https://github.com/lchen001/HAPI. The terms of use for most ML APIs (see, e.g., https://cloud.google.com/terms and https://azure.microsoft.com/en-us/support/legal/) require no sublicensing to a third party. They do not prevent evaluating and analyzing their performance. In fact, evaluating and comparing the performance of different cloud services is not only desired by users but also encouraged by cloud providers. For example, Google Cloud provides its own performance measurement tool (https://cloud.google.com/free/docs/measure-compare-performance#:~:text=Google%20Cloud%20Platform%20provides%20two,%2Dto%2Ddate%20and%20unbiased). This is probably because a systematic study of the ML APIs can help the providers improve their services. For example, gender shade [30], the seminal work analyzing bias and stereotypes embedded in face detection APIs, has helped ML API providers improve their services and thus been appreciated by the industry. We have added a detailed discussion for this in the appendix (see line 614 - line 626, page 19).

---

> > ### Author Response · Authors · 2022-08-19
> > **Thank you very much for your thoughtful review! We have updated the paper to incorporate your feedback. (2/2)**
> >
> > ***[What are the selection criteria of the datasets?]***: We chose existing datasets for a few reasons. First, the ML community is familiar with the datasets and they are relatively well annotated and evaluated. Second, those datasets can be easily assessed on the internet. Third, those datasets covered a diverse range of real-world scenarios (for example, the COCO dataset included objects in outdoor/indoor environments, at a small/large scale, and with different brightness). In fact, based on our conversation with many practitioners, there is a large interest in understanding commercial APIs’ performance on those datasets. Thus it is a good starting point to evaluate ML APIs on those popular datasets. We have added a detailed discussion in the appendix (see line 654 - line 660, page 21).
> >
> > ***[Do you plan to add more datasets?]***: Yes. As part of the maintenance and development plan, more datasets, emerging ML APIs for more tasks will be added to HAPI. We have added a detailed discussion in the appendix (see line 627 - line 639, page 19).
> >
> > Thank you again for your feedback, which has improved our paper! We hope you would consider increasing your score in light of our detailed response and revision. Thank you!

---

### Review · Ethics_Reviewer_CdrF · 2022-08-22

**Recommendation:** 1

**Ethics Review:**

In terms of the strengths of this submission I would like to highlight that section 4 does not only focus on metrics related to accuracy, but also considers important ethical concerns: the fairness of predictions.

The reviewers have raised several ethical concerns about this submission. However, as stated by Reviewer V17d, the authors' response addressed most of the ethical issues related to this submission:

> In light of the author's R&R to address concerns - especially with regards to declaring funding sources, which was one of the fundamental issues raised in my review - and "plan to enlarge the set of ML APIs, datasets" to alleviate any perceived selection bias, I am pleased to revise my score as above.

Still, one important point was not addressed. Reviewer 82x1 noted that
> It is unclear to me whether the FER+, EXPW, RAFDB and AFNET facial emotion recognition datasets have been collected with consent of the human subjects.

The authors' response did not address this concern
> [Have the FER+, EXPW, RAFDB and AFNET facial emotion recognition datasets been collected with the consent of the human subjects?]: Thank you for the question. All of these are public benchmark datasets that are widely used in the machine learning community. For example, the RAFDB paper has been cited more than 700 times in the past five years. We did not collect any of these image data ourselves. We wanted to use existing public datasets because they have been frequently used to benchmark algorithms (though not the commercially available ML APIs that we studied here).

Even though these datasets were not gathered by the authors, and even though they may be widely used in the ML community, it would be important to comment on potential ethical issues associated with them, such as Reviewer 82x1 concerns about how the data was collected and whether the human subjects have given their consent.

---

Update Aug 23:
I would like to thank the authors for addressing this issue. I have updated my score accordingly.

---

> ### Author Response · Authors · 2022-08-23
> **Thank you for the thoughtful ethics review and we have incorporated your feedback into our paper**
>
> Thank you for the thoughtful ethics review and suggestions. We have added the following discussion to the revised paper (see line 666 - line 677, page 21 - page 22):
>
> **"According to the original documents of the facial emotion datasets (FER+ [25], RAFDB [55], EXPW [71], and AFNET [60]), all the face images in these four facial emotion datasets were collected via querying search engines (e.g., Google, Bing, and Yahoo!) with certain keywords (e.g., happy faces). While the images are publicly retrievable from search engines, we did not find clear documentation of the individual consent process for these datasets. We recognize that facial photos are sensitive data, and will remove photos from HAPI upon request. Moreover, photos curated online may not fully represent the general public, and emotion annotations can be subjective and noisy. Therefore, the analysis of these datasets should be interpreted with care. For example, the fact that an API’s performance on some of these datasets changes over time is important to know, while the absolute performance across different datasets may not be directly comparable. We will continue to work with the machine learning community to expand HAPI to include high-quality benchmark datasets."**
>
> We agree with you that an important application of HAPI is to quantify disparity across subgroups in the commercial ML APIs. We have further emphasized this point as a valuable research direction for the ML community.

---

### Author Response · Authors · 2022-08-19
**Thank you for your thoughtful reviews! Summary of our major updates.**

We thank all reviewers for their thoughtful reviews. We have carefully updated the paper to incorporate all of their feedback (which can be found in the updated files). Here, we summarize the major updates:

***[New evaluations in August 2022]***: We have added evaluation of all structured prediction APIs in August 2022. Specifically, these new collections include ML API predictions on all datasets for multi-label image classification (PASCAL, MIR, COCO), scene text recognition (MTWI, ReCTS, LSVT), and named entity recognition (CONLL, GMB, ZHNER). Interesting shifts were observed compared to six months ago (Feb/March 2022). This adds more frequently updated data to HAPI. We have added a discussion to the main paper (see line 247- line 253 and Figure 5, page 9) and the appendix (see line 640 – line 653 and Figure 7 and Figure 8, page 19 - page 21).

***[The maintenance and development plans for HAPI]***: We have included a detailed plan for updating and expanding HAPI going forward (see line 627 - line 639, page 19). We also believe that the current HAPI dataset is already a valuable resource as it is the first large-scale evaluation of commercial ML APIs and contains data from more than two years.

***[The selection criteria of ML APIs and datasets.]***: We explain how our choices of ML APIs and datasets were appropriate, as these datasets and APIs are (i) popular in the ML community, (ii) easily accessible online, and (iii) represent a diverse range of applications and companies. We have added a detailed discussion in the appendix (see line 654 - line 665, page 21).

We provide more details in our response to each reviewer. The revision has significantly strengthened the paper. Please let us know if you have any further questions and we are happy to follow up!

---

### Meta-Review · Area_Chair_mHir · 2022-09-08

**Recommendation:** Accept
**Confidence:** 4

**Metareview:**

This paper proposes a dataset of commercial ML API performance through time, which is an interesting topic given the wide usage of such ML APIs, and their evolving performance (and blindspots) isn't necessarily always considered. The initial reviews had high variance: There were some very high scores, but also recommendations for rejection. There were some very valid concerns raised by the reviewers: whether or not the use of popular pre-existing standard benchmarks increases the risk of overfitted performance, the coverage of APIs and datasets considered in API, whether yearly measurements is too sparse, and if the currently collected data represents a long enough period of time (unfortunately, I don't think it'd be possible, but it'd be particularly interesting to see earlier history). These are considerations that I encourage the authors to consider moving forward, as it appears there are plans for further expansion.

After some active discussion and revisions from the authors, however, the reviewers have converged toward recommending acceptance. Given the importance of MLAAS today and likely the future, datasets such as HAPI can have wide applicability for measuring performance, weaknesses, bias, fairness, and other model characteristics. As such, I recommend acceptance.

---

### Decision · Program_Chairs · 2022-09-16

Accept